# Seasonal Variability of the Oxygen Minimum Zone off Peru in a
# high-resolution regional coupled model
**O. Vergara[1], B. Dewitte[1], I. Montes[2], V. Garçon[1], M. Ramos[3,4,5], A. Paulmier[1], and O.**
**Pizarro[6,7]**
[1]{Laboratoire d'Études en Géophysique et Océanographie Spatiales, CNRS/CNES/UPS, UMR5566,
Toulouse, France}
[2]{Instituto Geofísico del Perú (IGP), Lima, Perú}
[3]{Departamento de Biología, Facultad de Ciencias del Mar, Universidad Católica del Norte,
Coquimbo, Chile}
[4]{Millennium Nucleus for Ecology and Sustainable Management of Oceanic Islands (ESMOI),
Coquimbo, Chile}
[5]{Centro de Estudios Avanzado en Zonas Áridas (CEAZA), Coquimbo, Chile}
[6]{Department of Geophysics, University of Concepcion, Chile}
[7]{Millennium Institute of Oceanography, Chile}
Correspondence to: O. Vergara (oscar.vergara@legos.obs-mip.fr)

**Abstract**

In addition to being one of the most productive upwelling systems, the oceanic region off Peru is embedded in one of the most extensive Oxygen Minimum Zones (OMZs) of the world ocean. The dynamics of the OMZ off Peru remain uncertain, partly due to the scarcity of data and to the ubiquitous role of mesoscale activity on the circulation and biogeochemistry. Here we use a high-resolution coupled physical/biogeochemical model simulation to investigate the seasonal variability of the OMZ off Peru. The focus is on characterizing the seasonal cycle in Dissolved $O_2$ (DO) eddy flux at the OMZ boundaries, including the coastal domain, viewed here as the eastern boundary of the OMZ, considering that the mean DO eddy flux in these zones has a significant contribution to the total DO flux. Along the coast, despite the increased seasonal low DO water upwelling, the DO peaks homogeneously over the water column and within the Peru Undercurrent (PUC) in austral winter, which results from mixing associated with the increase in both the intraseasonal wind variability and baroclinic instability of the PUC. The coastal ocean acts therefore as a source of DO in Austral winter for the OMZ core, through eddy-induced offshore transport that is also shown to peak in Austral winter. In the open ocean, the OMZ can be divided vertically into two zones: an upper zone above 400m where the mean DO eddy flux is larger on average than the mean seasonal DO flux, and varies seasonally, and a lower part where the mean seasonal DO flux exhibits vertical-zonal propagating features that share similar characteristics than those of the energy flux associated with the annual extra-tropical Rossby waves. At the OMZ meridional boundaries where the mean DO eddy flux is large, the DO eddy flux has also a marked seasonal cycle that peaks in austral winter (spring) at the northern (southern) boundary. In the model, the amplitude of the seasonal cycle is 70% larger at the southern boundary than at the northern boundary. Our results suggest the existence of distinct seasonal regimes for the ventilation of the OMZ by eddies at its boundaries. Implications for understanding the OMZ variability at longer timescales are discussed.

# 1 Introduction

In addition to hosting one of the most productive upwelling systems, the South Eastern Pacific (SEP) is home to one of the most extensive Oxygen Minimum Zones (OMZs) of the world ocean (Fuenzalida et al., 2009; Paulmier and Ruiz-Pino, 2009). These oxygen deficient regions are key to understanding the role of the ocean in the greenhouse gases budget and in climate, and in the presently unbalanced nitrogen cycle (Gruber, 2008). The OMZs represent a net nitrogen loss to the atmosphere in the form of $N_2O$ (particularly the SEP OMZ: Farías et al., 2007; Arévalo-Martínez et al., 2015), in addition with other toxic or climatically-active gases, such as $H_2S$ and $CH_4$, respectively, in extremely low dissolved oxygen (DO) concentrations (Libes, 1992; Law et al., 2013). They might even limit the ocean carbon sequestration and act as $CO_2$ sources for the atmosphere (Paulmier et al., 2008; 2011). Furthermore, the OMZs contribute to the habitat compression of marine organisms, in a zone that sustains 10% of the world fish catch (Prince and Goodyear, 2006; Chavez et al., 2008). Therefore, understanding the dynamics behind the OMZ becomes not just a matter of scientific interest, but also a major societal concern.

In general, these low oxygen regions are considered to result from the interaction of biogeochemical and physical processes (Karstensen et al., 2008). The SEP presents high biological productivity, inducing a significant DO consumption mainly through the remineralization associated with a complex nutrient cycle supported by the intense upwelling. In addition, the SEP encompasses a so-called 'shadow zone', a near stagnant/sluggish circulation region next to the eastern basin boundary, not ventilated by the basin scale wind driven circulation (Luyten et al., 1983). Assuming a steady state, lateral oxygen fluxes from subtropical water masses and diapycnal mixing are expected to balance the oxygen consumption (Brandt et al., 2015). However, the diversity of environmental forcings in the SEP, and the variety of timescales at which they operate (Pizarro et al., 2002; Dewitte et al., 2011; 2012) has eluded a proper understanding of the processes controlling the OMZ structure and variability. On the one hand, the scarcity of data and rare surveys have only permitted to document the DO temporal variability at a few locations (e.g. Morales et al., 1999; Cornejo et al., 2006; Gutiérrez et al., 2008; Llanillo et al., 2013). On the other hand, the highly complex interaction between physical and biogeochemical mechanisms makes modeling and prediction of OMZ location, intensity and its temporal variability a challenging task (Karstensen et al., 2008; Cabré et al., 2015). Low resolution CMIP class coupled models still have severe biases of physical and biogeochemical origins,

particularly in Eastern Boundary Current systems (Richter, 2015), which has eluded the interpretation of long term trends in OMZ (Stramma et al., 2008; 2012; Cabré et al., 2015). Regional coupled biogeochemical modeling nonetheless has provided a complementary approach to gain insight in the dynamics of OMZ and its relationship with climate (Resplandy et al., 2012; Gutknecht et al., 2013a). One recent modeling effort in understanding the dynamics behind the OMZ in the Eastern Tropical Pacific comes from Montes et al. (2014). This study provided a first regional simulation of the OMZ in the SEP, and summarized the elements involved in maintaining the OMZ found off the coast of Peru as resulting from a delicate balance between (i) the equatorial current system dynamics: the relatively oxygen-rich waters carried by the Equatorial Undercurrent (EUC), the relatively oxygen poor and nutrient rich waters carried by the primary and secondary Tsuchiya Jets (primary and secondary Southern Subsurface Countercurrents, pSSCC and sSSCC, respectively), and (ii) the high surface productivity rates induced by the coastal upwelling, which in turn triggers an intense oxygen consumption in the subsurface. Their model experiments also showed that different Eddy Kinetic Energy (EKE) levels, induced by different representations of the mean vertical structure of the coastal current, may contribute to expand or erode the upper boundary of the OMZ.

The study by Montes et al. (2014) established a benchmark in terms of numerical modeling of the OMZ in the SEP, focusing on its permanent regime and connection with the equatorial current dynamics. In the present study, we also take advantage of the regional modeling approach in order to investigate the mechanisms associated with the seasonal cycle of DO within the OMZ. The motivation for focusing on seasonal variability is three-folds: 1) A better knowledge of the processes acting on the OMZ at seasonal timescale is viewed as a prerequisite for interpreting longer timescales of variability (ENSO, decadal); 2) the scarcity of quality long term subsurface biogeochemical data in the SEP is a limitation for tackling the investigation of OMZ variability at low frequency; 3) To the authors' knowledge, this issue has not been addressed in the literature for the Eastern Tropical Pacific, although it has been a concern for other tropical oceans (Resplandy et al., 2012; Gutknecht et al., 2013a; Duteil et al., 2014).

Here, besides investigating to which extent the seasonal OMZ variability can relate to the variability of the environmental forcing in the SEP (local wind, equatorial Kelvin and extra-tropical Rossby waves), our interest is on examining the DO budget (i.e. the balance between oxygen sources and sinks) and relating it to the physical DO flux. In particular, since the Peruvian region is the location of a relatively intense eddy activity (Chaigneau et al., 2009), the question of whether or not eddy activity is involved

in the seasonal variability of the OMZ arises, and calls for assessing its contribution to the DO flux.
There is growing evidence that mesoscale activity has a key role on the biogeochemical cycles and the
OMZ structure in EBUS (Duteil and Oschlies, 2011, Nagai et al., 2015). Most studies addressing the
role of mesoscale processes in the OMZs have focused on the ventilation from the coastal domain,
where the primary production bloom provides nutrients and DO anomalies that are in turn transported
offshore (Stramma et al., 2013; Czeschel et al., 2015; Thomsen et al., 2016). Gruber et al. (2011)
showed that mesoscale activity is prone to reduce the biological production and offshore carbon export
in upwelling systems by both rectifying on the mean circulation (i.e. eddy-induced mixing tends to
flatten the isotherms nearshore and reduce the upwelling) and changing its nutrient transport capacity.
This process has been to some extent supported by observations in the Peruvian OMZ (Stramma et al.,
2013). In this sense, the mesoscale activity represents a ventilation pathway for the OMZ, through the
offshore transport of oxygen-enriched waters. The ventilation of the OMZ could also take place at its
meridional boundaries where strong mean DO gradients are found along with eddy activity. Recently,
Bettencourt et al. (2015) proposed that mesoscale eddies shape the Peruvian OMZ by controlling the
diffusion of DO into the OMZ at the meridional boundaries. Although it is likely that both processes
are important for understanding the OMZ structure, it has not been clarified to which extent the
variability of the OMZ could be understood in terms of the changes in the DO eddy flux into the OMZ
through these different pathways. The mesoscale activity also exhibits a significant meridional
variability off Peru (Chaigneau et al., 2009), which questions if the offshore ventilation process can
operate effectively for modulating the whole OMZ. Another related open question is at which
timescales the ventilation process through eddies-induced mixing can operate effectively. In this paper
we will tackle these issues from a regional modeling approach, focusing on the seasonal timescale.
The paper is organized as follows. After the Introduction (Section 1), we detail the observations and
model configuration used in the study, as well as the methodology employed in the treatment of the
information (Section 2). We also evaluate the realism of the simulation against the available
observations in reproducing the main characteristics of the OMZ. The subsequent section (Section 3)
characterizes the DO annual cycle inside the OMZ. Section 4 opens with the analysis of the seasonal
variability of the coastal OMZ, and the contribution of the DO budget terms associated with it. This
analysis is followed by the results on DO flux directed offshore and completed by the analysis of DO
flux across the OMZ meridional boundaries. Section 5 presents a discussion of the main results and

1 Section 6 presents a summary and the concluding remarks.

## 2 Data description and Methods

### 2.1 Data

**Dissolved Oxygen concentration from CARS**

The CSIRO Atlas of Regional Seas (CARS) is a climatological product derived from a quality-controlled archive of historical subsurface ocean measurements, most of which was collected during the past 50 years (Additional information might be found in the website of the project: http://www.marine.csiro.au/~dunn/cars2009/). For the present study, we use the CARS2009 version of the CARS product (Ridgway et al., 2002), which has an horizontal resolution of 0.5°x0.5° and 79 vertical levels, with a 10m resolution near the surface layer. We use CARS to assess the model skills in simulating the OMZ mean state and variability. One advantage of this product is its refined interpolation treatment near steep topography, in comparison to other products such as the World Ocean Atlas (Dunn and Ridgway, 2002). Also, it includes the annual and semiannual oxygen cycles, although the semiannual cycle is available only for the first 375 m over the region of interest due to the scarcity of data.

**Chlorophyll-a concentration from SeaWiFS**

SeaWiFS 8 day composites at 0.5°x0.5° resolution chlorophyll product (version 4), between January 2000 and December 2008, is used to compute the surface chlorophyll seasonal cycle (McClain et al., 1998; O'Reilly et al., 2000).

**Sea Surface Temperature (SST)**

The NOAA Optimum Interpolation SST (OISST V2) product, is contrasted against the simulation SST. This product is an analysis constructed by combining observations from different platforms (satellites, ships, buoys) on a regular global grid. More information about the methodology used to construct this data set may be found in Reynolds et al. (2007), and the product website (https://www.ncdc.noaa.gov/oisst). The version used in this study corresponds to daily SST maps with a spatial resolution of 0.25°x0.25°, spanning the period 2000-2008.

**Sea Level Height (SLH)**

TOPEX/JASON1 merged SLH data set, distributed by the Sea Level Research Group, University of Colorado (http://sealevel.colorado.edu/) is used to derive the geostrophic velocity field and the mean

EKE field. This data set corresponds to a globally-gridded 0.25°x0.25° weekly product. The
information used corresponds to the period 1993-2008. Further details on this product may be found in
Nerem et al. (2010).

## 2.2 Model simulation

We use a high resolution simulation of the South Eastern Pacific, based on the hydrodynamic model
Regional Ocean Modeling System (ROMS) circulation model (see Shchepetkin and McWilliams,
2005; 2009 for a complete description of the model) coupled with a nitrogen-based biogeochemical
model developed for the Eastern Boundary Upwelling Systems (BioEBUS, Gutknecht et al., 2013ab),
hereby referred as CR BIO.
The model is used at an eddy-resolving resolution (1/12° at the equator) for a region extending from
12°N to 40°S and from the coast to 95°W -nevertheless this study only focuses on the domain spanning
the latitudes of Peru and Ecuador (Fig. 1)- with lateral open boundaries at its northern, southern and
western frontiers. The physical model resolves the hydrostatic primitive equations with a free-surface
explicit scheme, and a stretched terrain-following sigma coordinates on 37 vertical levels. The
configuration is similar to Dewitte et al. (2012), that is the open boundary conditions are provided by
3-daily mean oceanic outputs from SODA (Version 2.1.6) for temperature, salinity, horizontal velocity
and sea level for the period 1958-2008, while wind stress and speed forcing at the air/sea interface
come from the NCEP/NCAR reanalysis. The atmospheric fields have been statistically downscaled
following the method by Goubanova et al. (2011) in order to correct for the unrealistic wind stress curl
near the coast of the NCEP Reanalysis (see Cambon et al. (2013) for a validation of the method for
oceanic applications). Atmospheric fluxes were derived from the bulk formula using the temperature
from COADS 1°x1° monthly climatology (daSilva et al., 1994). Relative humidity and short wave and
long wave radiations are also from COADS. Bottom topography is from the GEBCO 30 arc-second
grid data set, interpolated to the model grid and smoothed as in Penven et al. (2005) in order to
minimize the pressure gradient errors and modified at the boundaries to match the SODA bottom
topography. This model configuration has been validated from satellite and in situ observations in
Dewitte et al. (2012) with a focus on mean state interannual variability. In general the model is skillful
in simulating the mean SST field (Fig. 1ac) as well as other main aspects of the mean circulation (e.g.
Peru Chile Undercurrent, EKE, see Figure 3 in Dewitte et al. (2012)), although with a slight cold bias
(~1°C), that could be partly attributed to the use of climatological heat flux forcing (Fig. 1d). The cold
bias observed here could be also due to a systematic warm bias in the AVHRR Pathfinder data,
observed in the nearshore regions where high SST gradients exist, specifically in the Humboldt region
(cf. Dufois et al., 2012). This product is extensively used in the construction of the OISST dataset
(Reynolds et al., 2007).
The mesoscale activity diagnosed from the mean EKE, has a comparable pattern than altimetry,
although with a larger amplitude (Fig. 1ef). Similar levels of mesoscale activity have been obtained by
previous modeling studies in the Peruvian region (e.g. Echevin et al., 2011; Colas et al., 2012).
The ocean model within this configuration is coupled to the BioEBUS model following similar
methodology than Montes et al. (2014). BioEBUS uses two compartments of phytoplankton and
zooplankton, small (flagellates and ciliates, respectively) and large (diatoms and copepods,
respectively), detritus, dissolved organic nitrogen and the inorganic nitrogen forms nitrate, nitrite and
ammonium, as well as nitrous oxide (see Gutknecht et al., 2013ab, for a description of the model). The
open-boundary conditions for the biogeochemical model are provided by the climatological CARS data
set (nitrate and oxygen concentrations) and by SeaWiFS archive (chlorophyll-a concentration).
Additional biogeochemical tracers are computed following Gutknecht et al. (2013ab). Initial
phytoplankton concentration is defined as a function of vertically extrapolated satellite Chl-a following
Morel and Berthon (1989). An offshore decreasing cross shore profile, following in situ observations, is
applied for zooplankton, and a vertical constant (exponential) profile is used for detritus (nitrite,
ammonium and dissolved organic nitrogen), respectively. In order to get a realistic solution for the
region, the model parameters were tuned to simultaneously fit modeled oxygen and nitrate fields to
observations (see Table A1 of Montes et al. (2014) for parameter values). These changes were
motivated by the need to adjust the microbiological rates to values observed in the SEP. Within this
parameter configuration, BioEBUS has been shown to be skillful for simulating the OMZ off Peru
(Montes et al., 2014). In particular the pattern correlations between the model and observations for both
the annual mean and the seasonal cycle inside the OMZ present comparable scores (>0.85, cf. Montes
et al. (2014)) as well as low standard deviations (i.e. in the order of the observed values). The model
was run over the period 1958-2008 with a 10-year spin-up obtained by repeating the year 1958.
Although, after the spin-up, the simulation has reached stable conditions and the OMZ volume does not
drift, we focus in the present study only on the period 2000-2008.
The reason for focusing on the last ten years of our simulation is also motivated by the fact that the
atmospheric momentum forcing is close to the satellite QuickSCAT winds by construction (see
Goubanova et al. (2011) for details) so that this period of the simulation is the one when the model is
the most constrained by observations. Most previous modeling studies for this region (Penven, et al.
2005; Montes et al., 2010, 2014; Echevin et al., 2011; Colas et al., 2012) have also used a wind forcing
from the QuickSCAT scatterometer, which provides a benchmark for assessing our simulation.
A monthly mean climatology is calculated for all variables over this period from the 3-day mean
outputs of the model, which can be compared to the CARS data.
Consistently with Montes et al. (2014), the coupled simulation is skillful in simulating the mean
characteristics of the OMZ off the Peruvian coast (Figures 2 and 3). In particular the thickness and
location of the model OMZ core limits are realistic, and in good agreement with previous studies (Fig.
2; e.g. Paulmier et al., 2006; Cornejo and Farías, 2012; Montes et al. 2014). Note that the simulation
reproduces a thinner OMZ around 10°S in comparison to CARS, which agrees with the results
obtained by Montes et al. (2014), (see Fig. 2 in that study). Close to the western boundary of our model
domain, the simulated OMZ also exhibits a realistic vertical structure (Fig. 3) with comparable
concentration in DO than observations in the vicinity of the Equatorial Undercurrent (~100 m;
Equator). Furthermore, the simulation is consistent in reproducing the oxygen consuming processes, as
supported by the Apparent Oxygen Utilization (AOU; Fig. 4), also in good agreement with previous
studies (cf. Figure 8 in Cabré et al. (2015)). AOU was computed as the difference between the DO
concentration and the saturated oxygen ($O_2$sat) concentration (AOU=$O_2$sat-$O_2$) with $O_2$sat computed
following the methodology of García and Gordon (1992). The realistic representation of the oxygen
consuming processes is reflected by the Particulate Organic Carbon flux as well (Fig. 5a), whose
values at 100m fall within the observed range for the region (30-60 gC m$^{-2}$ yr$^{-1}$ in the shelf area; Dunne
et al., 2005; Henson et al., 2012). In addition, the low transfer efficiency of carbon (10-15% or lower
over and next to the shelf; Henson et al., 2012), from the euphotic zone to greater depths (Fig. 5b),
implies that the remineralization processes take place at realistic depths, and therefore allow for a
correct vertical representation of the OMZ (cf. Fig. S2 in Cabré et al. (2015) for comparison).
The core of the OMZ, defined with a suboxic concentration ([DO] < 20 $\mu$M; µM will be used to refer to
µmol L$^{-1}$ in all the text and Figures), occupies nearly 23% of the domain volume (Fig. 6a), with the less
oxygenated layers comprised between 5°S and 15°S, and 100 m and 600 m depth (Fig. 3). As expected,
the simulation presents more details than the climatological product (Fig. 3). Moreover, we computed a

geographical OMZ overlapping metric following Cabré et al. (2015), which quantifies the spatial agreement of the OMZ volume distribution between the simulation and CARS, varying between 0 (no agreement) and 1 (perfect collocation). We obtained a value of 0.79, which is ~58% above the best CMIP5 models used in Cabré et al. (2015).

Despite the overall good agreement between the model and observations, the modeled oxygen content is however underestimated as compared to CARS in certain regions of the domain, particularly southwards of 20°S (Fig. 3a) and close to the coast (Fig. 3d). In particular, the volume of the suboxic DO concentration range is 6% lower in the simulation (Fig. 6a), which is comparable to the differences obtained by Montes et al. (2014) in a similar analysis of the OMZ.

The modeled DO distribution is also characterized by finer spatial scales of variability inside the OMZ compared to observations (Figures 3c and 3d). In particular, the model oxycline is shallower and with a more intense DO gradient than the observations, which has been also observed in a simulation of the Arabian Sea OMZ (Resplandy et al., 2012), suggesting that the CARS data set may not have a sufficient vertical resolution to realistically resolve the oxycline at some locations. Also, it must be kept in mind that CARS is built using all the available data from the second half of the twentieth century (1940-2009), whereas we focus on the period 2000-2008 for the simulation, which is known to be a colder period than the previous decades in the eastern tropical Pacific (Henley et al., 2015). Other limitation for the comparison between model and data includes the errors associated with the scarcity of data in some regions (Bianchi et al., 2012). Nonetheless, the simulation is in good agreement with CARS in terms of mean characteristics of the OMZ, as well as the mean oxygen concentration and its distribution (Figures 4, 6a).

In order to evaluate the realism of the seasonal cycle, we estimate the seasonal variability of the volume of water within the suboxic DO concentration range 0-20 $\mu M$ in both the model and data (Fig. 6b). The results indicate that, despite a weaker amplitude (by 15% on average), the seasonal cycle of the OMZ core is relatively well simulated by the model. For hypoxic DO volume in the range 40-50 µM, the agreement is as good as inside the OMZ core, with a Pearson correlation value of 0.9 and a volume RMS difference of 16%, between the simulation and the observations.

In order to summarize the model validation, we present a Taylor diagram showing the statistics of the comparison between the model and observations for a depth range encompassing the OMZ (Fig. 7). This analysis indicates that within the present model configuration, we reach a comparable skill than

the model configuration of Montes et al., (2014) (their Figure 1). The good agreement of the seasonal
cycle between CARS and the simulation, in addition to the consistency of our results with those of
Montes et al. (2014), provides confidence in using the model outputs for investigating the processes
associated with the seasonal variability of the OMZ.
**2.3  Methods**
In this work, our approach is twofold: First, the biogeochemical processes for DO are investigated
explicitly through the on-line oxygen budget (1). Although this methodology can provide a direct
estimate of the seasonal variability in advection and mixing, it does not allow for a direct estimate of
the eddy contribution to DO change that can also vary seasonally. The DO flux associated with
different timescales of variability is therefore estimated. This consists in computing the temporal
average of the cross-products between DO and velocity anomalies. Anomalies can refer either to
seasonal anomalies and in that case, this provides the mean seasonal DO flux ( $\langle \tilde{u} \cdot \tilde{O}_2 \rangle$ , where ~ refers
to the seasonal anomalies), or to the intraseasonal anomalies (calculated here as the departure from the
monthly mean) and in that case, this provides an estimate of the mean DO eddy flux ( $\langle u' \cdot O_2' \rangle$ , where
the apostrophe refers to the intraseasonal anomalies). In this paper we are also interested in the
seasonality of the DO eddy flux. This is estimated from the monthly-mean seasonal cycle of the mean
DO eddy flux calculated over a 3-month running window, and is now referred to as $\overline{\langle u' \cdot O_2' \rangle}$ . The
climatological EKE activity is estimated similarly.
The DO budget consists in the following Equation:
$$\frac{\partial O_2}{\partial t} = -\vec{u} \cdot \left( \vec{\nabla} O_2 \right) + K_h \nabla^2 O_2 + \frac{\partial}{\partial z} \left( K_z \frac{\partial O_2}{\partial z} \right) + SMS \left( O_2 \right) \quad . \tag{1}$$
The first three terms on the right hand side represent the physical processes involved in the changes in
oxygen concentration. The first term stands for the advection of oxygen, with $\vec{u}$ the velocity vector
(note that the model determines the vertical velocity component from the continuity equation). The
second term corresponds to the horizontal subgrid-scale diffusivity (with $K_h$ the eddy diffusion
coefficient equal to 100 $m^2 s^{-1}$ in this version of the model), and the third term corresponds to the
vertical mixing (with turbulent diffusion coefficient $K_z$ calculated based on the KPP mixing scheme
(Large et al., 1994)). Note that the model has also numerical diffusion associated with inherent spurious
diapycnal mixing of the numerical scheme, so that $K_h$ is empirically adjusted
The fourth term represents the "Sources-Minus-Sinks" contribution to the oxygen changes, directly due
to biogeochemical activity. Biogeochemical processes correspond to the sum of oxygen sources and
sinks, namely the photosynthetic production, and the aerobic processes (oxic decomposition, excretion
and nitrification). In this study, for simplicity, those will be considered as a summed-up contribution to
the DO rate of change, whereas physical processes will be divided into advection and mixing terms.
Each term of this oxygen budget is determined on line at each time integration. While horizontal
diffusion and vertical diffusivity are explicit sources of mixing, they are not the only terms contributing
to mixing. Later on in the paper, unless stated otherwise, the term mixing will refer to the integrated
effect of all processes contributing mixing directly or indirectly. Besides the horizontal diffusion (
$K_h \cdot \nabla^2 O_2$ ) and vertical mixing ( $\frac{\partial}{\partial z}\left(K_z \frac{\partial O_2}{\partial z}\right)$ ), mixing can be also induced by non-linear advection.
The latter corresponds to $\left(u' \partial u'/\partial x\right)+\left(v' \partial v'/\partial y\right)+\left(w' \partial w'/\partial z\right)$ , assuming the Reynolds decomposition
for the velocity field, i.e. $\vec{u}+\vec{u}'$ , where $\vec{u}'$ accounts for the intraseasonal variability (periods lower
than ~3 months).
In the SEP, the subthermocline seasonal variability can be interpreted as resulting from the propagation
of Extra-Tropical Rossby Waves (ETRW). ETRW radiate from the coast and propagate vertically,
inducing a vertical energy flux, whose trajectory follows the theoretical Wentzel-Kramers-Brillouin
(WKB) ray paths (Dewitte et al., 2008; Ramos et al., 2008). The energy flux results from the phase
relationship between vertical velocity associated with the vertical displacement of the isotherms, and
the pressure fluctuations associated with them. In the regions sufficiently below the thermocline for
DO consumption to become weak (that is DO can be considered a passive tracer), it is expected that
changes in DO relate to the anomalous velocity field, and that the DO flux shares comparable
characteristics than the Eliassen-Palm flux (EP flux; Eliassen and Palm, 1960). The trajectories of the
WKB ray paths are a function of latitude, local stratification and the phase speed of the Rossby wave
(see Ramos, et al., (2008)). The latter consists in the superposition of a certain number of baroclinic
modes, in order to propagate vertically, so the phase speed can range from values between $c_1$ to $c_n$,
where $c_n$ is the theoretical phase speed of a $n^{th}$ baroclinic mode, obtained from the vertical mode
decomposition of the local density profile.
**3 Characteristics of the DO annual cycle**
While the annual signal is a conspicuous feature inside the region (Fig. 6b), it could manifest
differently across the OMZ. As a first step towards investigating processes driving the rate of DO
change, it appears important to document the vertical structure variability of the DO annual cycle
within the OMZ. The amplitude and phase of the annual harmonic of the model DO climatology is
presented along a zonal section off central Peru (12°S, Fig. 8ab), where the OMZ core is extensive
(Fig. 2). The DO climatology has been normalized by its RMS (Root Mean Square), in order to
emphasize the regions where the amplitude in DO changes (and mean DO) is weak. The amplitude
reveals a complex pattern with three regions of large relative variability: 1) near the coast (i.e. fringe of
~150 km) between the oxycline and 400 m; 2) offshore between 82°W and 84°W in the upper 400 m
and 3) below 500 m. The phase lines over these three regions suggest distinct propagating
characteristics: whereas in the coastal region there is no propagation, in the offshore and deep region,
there is indication of a westward propagation. In the region below 500 m, the phase lines tend also to
be parallel and slope downward, suggestive of westward-downward propagation (estimated phase
speed of ~2.5 cm s$^{-1}$). These propagating characteristics can be evidenced in the Hovmöllers of the
recomposed annual cycle at the depth of 150 m (Fig. 8c) and 700 m (Fig. 8d). While at 150 m the
annual signal does not clearly propagate and only shows two domains of high amplitude, separated by
low amplitude values (Fig. 8c), there is a clear westward propagation of the DO anomalies at 700 m,
with the phase speed increasing westward. At 400 m, the propagation is only observed west of 81°W
(Fig. 8b). In addition to the large vertical structure variability of the annual cycle, the OMZ annual
cycle is also characterized by a large horizontal variability in particular at its northern and southern
boundaries. This is illustrated from Figure 9, that displays the amplitude of the annual cycle of the DO
climatology at 400 m, and evidences amplitude peaks at the OMZ meridional boundaries (between the
20 and 45 $\mu$M isopleths).
The annual variability pattern evidenced above results from a delicate balance between the physical
processes (namely advection and mixing, cf. Eq. (1)) and the biogeochemical processes (consumption
versus production). As a first step towards investigating each term of the DO budget, it is interesting to
evaluate the relative contribution of the physical and biogeochemical fluxes to the DO variability at
seasonal scale. The RMS of the climatological fluxes along a section at 12°S indicates that the
maximum amplitude of the seasonal fluxes takes place near the oxycline and along the coast over the
whole water column (Figure 10). The relative importance of the physical processes against the

biogeochemical processes varies across the OMZ. At the coast and near the oxycline, the annual

variability of the biogeochemical processes reaches values almost half those of the variability in

physical processes (Fig. 10c), as a consequence of the proximity to both the well lit and highly

productive part of the water column, and the high remineralization activity that occurs near the

oxycline. Towards offshore and at depth, the relative importance of the variability of the

biogeochemical processes reduces gradually. Near ~300 m the variability of the biogeochemical

processes is nearly 1/5 of the physical processes variability. Below ~300 m, and towards the lower part

of the OMZ core and below, the physical processes variability is one order of magnitude larger.

Consequently, the distribution of DO in the lower part of the OMZ is rather a function of

advection/diffusion than a consequence of the biogeochemical processes, although DO consumption

even at very low levels has the potential to generate local gradients and therefore induce advection. The

spatial heterogeneity in the seasonal DO changes induced by the biogeochemistry and dynamics as

described above, appears as an ubiquitous feature in the OMZ. To illustrate this, we estimate the

proportion of explained variance of the seasonal DO rate of change by the physical fluxes as:

$$R^2_{Phys.} = \left(1 - RMS\left(Biogeochemical\ Fluxes\right) / RMS\left(Total\ Fluxes\right)\right) \cdot 100 \ . \tag{2}$$

Figure 11ab presents the results of $R^2_{Phys.}$ at 100 and 450 m depth, which evidences that the relative

importance of the physical fluxes versus the biogeochemical fluxes in the seasonal DO variability

increases with depth, and is enhanced at the OMZ boundaries. On the other hand, the biogeochemical

fluxes explain more than 50% of the variance in seasonal DO change rate in a narrow (~ 200 km width)

coastal fringe that extends more offshore to the north of the domain (around 8°S; Fig. 11a) and

vertically down to 300 m (Fig. 11c).

Based on the above analysis, it is clear that the coastal region (first 200-300 km from the coast) below

the oxycline corresponds to a territory where the seasonal variability of biogeochemical and physical

fluxes have a comparable magnitude, whereas outside this region, notably in the lower part of the OMZ

core, the physical fluxes variability dominates over the biogeochemical fluxes variability at seasonal

timescale. Hereafter we examine the possibility of two distinct regimes of OMZ dynamics at seasonal

timescale: one associated with the upper OMZ (including coastal domain and meridional boundaries),

and the other one associated with the deep OMZ. In the following we investigate the processes

responsible for the DO flux.

## 4  Seasonality of the OMZ ventilation

It has been shown for the SEP that the DO content near the coast is set to a large extent from the transport of oxygen deficient waters from the equatorial current system, particularly the oxygen depleted sSSCC (Montes et al., 2014). Therefore, the seasonal variability of DO is likely to result in part from the seasonal variability of the different branches of the EUC in the far eastern Pacific. Local wind stress forcing (and its intraseasonal activity) has also a marked seasonal cycle off Peru (Dewitte et al., 2011) which may impact both the upwelling dynamics -through Ekman pumping/transport- and mixing. Some studies also argue that the DO exchange between the coastal domain and the OMZ takes place through the offshore transport of DO poor waters by eddies (Czeschel et al., 2011), implying that the variability of such processes is set up by coastal processes that determine the nature of the DO source. As a first step, we investigate the mechanisms responsible for the seasonal variability in DO along the coast, which can be considered as the eastern boundary of the OMZ. This is aimed at providing material for the interpretation of the offshore DO flux variability.

## 4.1  The coastal domain as the eastern boundary of the OMZ: variability and mechanisms

We analyze the seasonal variability along the coast, at a section at 12°S. Similar results are obtained for latitudes between 7°S and 14°S (not shown), which corresponds to the latitude range where the PUC is well defined. The results are also presented in terms of the first EOF mode, in order to ease the interpretation of the variability, reduced as a spatial pattern modulated by a seasonal timeseries. It was verified in particular that the consideration of the first EOF mode of each term leads to an almost perfect closure of the DO budget (see below, Table 1). Figure 12 displays the first EOF mode of various climatological fields in a section at 12°S near the coast and from the oxycline (45 $\mu M$ isoline) to the depth of 300 m. Figure 13 shows the principal components associated with the first EOF mode patterns. The seasonal DO cycle is dominated by an annual component, with a peak centered in August (Fig. 13a), and the largest variability at the coast below the oxycline that extends offshore and downward, resulting in an elongated tongue below 100 m near ~78°W (Fig. 12a). During the first quarter of the year, oxygen anomalies remain relatively stable (oxygen rate nearly zero, Fig. 13b), and negative, due to a high production of organic matter in Austral summer (cf. Fig. 1c of Gutiérrez et al., 2011) that stimulates a subsurface oxygen consumption associated with the degradation of this organic

matter. DO anomalies start to increase during the second quarter, become positive in June and reach
their maximum in August (Fig. 13a). The peak anomaly in Austral winter could be understood in terms
of the increased mixing (see Fig. 13a showing EKE peaking in July) associated with the increase in
baroclinic instability due to the seasonal intensification of the PUC from June. Note that the pattern of
the first EOF of the alongshore current coincides with the mean position of the PUC (see Fig. 12b), so
that seasonal variations of the PUC can be interpreted in terms of the variations in the vertical shear of
the coastal current system. Other processes that may explain the peak DO anomaly in Austral winter
includes the reduced productivity and downwelling that peaks in June (Fig. 13c), associated with
seasonal equatorial downwelling Kelvin wave.
The following investigates the tendency terms of the DO budget, in order to quantitatively interpret the
DO seasonal cycle near the coast. Given that the analysis is performed inside the 45 µM isopleth, the
biogeochemical flux term is largely dominated by the "Sinks" terms (aerobic processes; one order of
magnitude larger than "Sources"), driven by organic matter remineralization and zooplankton
respiratory metabolic terms (not shown). For clarity, the seasonal DO budget is presented synthetically,
from the first EOF mode of the climatological advection, mixing (horizontal and vertical diffusion) and
biogeochemical fluxes terms. Although this does not warranty a perfect closure, it eases the
interpretation. Note that the residual resulting from the difference between the first EOF mode of the
rate of DO changes and the summed-up contribution of all the other terms in Figure 13b is rather weak,
validating to some extent our approach (see also Table 1). First of all, we find that the largest amplitude
of the mode patterns is found near the coast and inside the mean PUC core (Figs. 12d to 12g). During
the first part of the year (January to May), positive advection anomalies are compensated by mixing
(horizontal and vertical diffusion), and maintain the rate of DO change relatively low (Fig. 13b; Table
1). Biogeochemical fluxes anomalies are positive during that period, associated with a positive
anomaly of primary production in the well lit surface layers, implied by the high chlorophyll-a values
(Fig .13c). A positive oxygen anomaly is sustained by the advection terms and the biogeochemical
terms, and is balanced out by the mean advection of low DO waters carried by the PUC (Montes et al.
2010; 2014), generating the relatively stable oxygen values (oxygen rate nearly zero).
From May, the rate of DO changes increases concomitantly with EKE (Fig. 13ab), followed one month
later by mixing (horizontal and vertical diffusion), whereas advection and biogeochemical fluxes
decrease. By June-July, the intensification in alongshore winds (Fig. 13c) starts to propel the coastal

upwelling, which has two compensating effects: on one hand it triggers photosynthesis in the lit surface layers (DO rate turns to positive values) and on the other, it uplifts low oxygen waters from the OMZ. The intraseasonal wind activity also starts to increase at that time (cf. Fig. 13c; see also Dewitte et al., 2011) which favors mixing, and so the downward intrusion of positive DO anomalies (note the deepening of the mixed layer in Fig. 13c). The overall effect is an increase in DO which leads to a peak anomaly in August. At that time, the DO rate drops sharply due to the strong subsurface DO consumption (Table 1) associated with aerobic remineralization of organic matter produced earlier in the season (DO rate moves sharply to negative values) and the high mixing that brings DO depleted waters from the subsurface into the deepened mixed layer. Note that this is consistent with the decrease in surface chlorophyll-a (Fig. 13c) and the interpretation proposed by Echevin et al. (2008) to explain the Austral summer minimum in surface chlorophyll-a observed off Central Peru.

This change to oxygen poor conditions combines with the natural decrease in oxygen production towards the end of the upwelling season and coincides with a restratification of the water column, which restricts the oxygenated waters near the surface (Echevin et al., 2008). This altogether contributes to maintain a negative DO rate inside the coastal OMZ, despite the increase in anomalous DO flux from the advective terms and (later on) biogeochemical processes towards the end of the year. As a result, oxygen returns to low values towards the end of the year.

**4.2  Offshore flux**

While the coastal OMZ variability is heavily constrained by the environmental forcings –coastal upwelling, coastal current system and local wind– due to the shallow oxycline there, the offshore OMZ, as embedded in the shadow zone of the thermohaline circulation, is somewhat insensitive to direct local forcing and rather experiences remote influence in the form of westward propagating mesoscale eddies (Chaigneau et al., 2009) and ETRW (Ramos et al., 2008; Dewitte et al., 2008). The influence of westward propagating mesoscale eddies on the OMZ translates as the transfer of coastal water properties towards the open ocean (DO included), while these properties are altered during transport due to physical-biogeochemical interactions (Stramma et al., 2014; Karstensen et al., 2015). Towards the end of their lifetime, hydrographic and biogeochemical anomalies carried by eddies are redistributed in the ocean (Brandt et al., 2015), linking the coast and the open ocean. Although most eddies genesis takes place near the coast (Chaigneau et al., 2009) and seasonal ETRW have a coastally forced component (Dewitte et al., 2008), we expect different characteristics of the seasonal variability

in DO between the coast and the open ocean, given that oxygen demand will change from one region to
the other. We also distinguish the mean DO flux associated with the annual component of the
circulation that represents the transport in DO associated with seasonal change in the large scale
circulation, and the annual variability of the eddy DO flux that corresponds to the annual changes in the
transport due to eddies. These two quantities are diagnosed at 12°S (Figs. 14 and 15). The DO has been
normalized by its climatological variability in order to emphasize variability patterns where DO is low.
**Mean seasonal flux**
We first document the mean DO flux associated with the annual component of the circulation. It
consists in the mean of the cross-product of the annual harmonics of the climatological velocity and
DO (Fig. 14a). The results indicate that the amplitude of the annual DO flux is maximum near the coast
and below ~400 m and it tends to be orientated westward-downward, following approximately the
trajectories of theoretical WKB paths for the annual period Rossby wave. Note that this is consistent
with the westward propagating pattern of DO below 400 m evidenced earlier (Fig. 8). As a consistency
check, we also estimated the annual energy flux vector in the (x,z) plan associated with a long extra
tropical Rossby wave, that is $\left( \langle p^{1yr} \cdot u^{1yr} \rangle, \langle p^{1yr} \cdot w^{1yr} \rangle \right)$ where the superscript denotes the annual
harmonics and the bracket the temporal average (Fig. 14b). The flux vector indicates vertical
propagation of energy at the annual period and the pattern of maximum flux coincides approximately
with the region of maximum amplitude of the mean seasonal DO flux. This suggests that the annual
ETRW is influential on the DO flux below ~400 m. This is interpreted as resulting from the advection
of DO by the ETRW since biogeochemical fluxes have much less influence on the DO rate of change
below 400 m (Fig. 10c) and the amplitude of the annual cycle of climatological DO eddy flux has a
much reduced amplitude below that depth (Fig. 15a) suggesting a reduced contribution of horizontal
and vertical diffusion to the DO budget. Note that the DO (Fig. 15a) was normalized prior to compute
the DO eddy flux in order to render both the analysis akin, and therefore contrast the flux associated
with the annual ETRW against the annual DO eddy flux. It was verified that the vertical structure
variability of the annual DO flux described above for the section of 12°S is comparable at other
latitudes within the OMZ. In particular the annual DO flux tends to remain homogeneous along
trajectories mimicking the energy paths of the ETRW at annual period which slope becomes steeper to
the South (not shown).
**Seasonal eddy flux**
As previously described, the annual amplitude of the climatological DO eddy flux is the largest in the
upper 400 m near the coast at 12°S consistently with the high EKE in this region. Since EKE is large
along the coast of Peru, exchange of DO induced by eddies could be expected at all latitudes, with a
direction that depends on the sign of the DO gradient at the coast. Figure 16 presents the annual
harmonic of the climatological DO eddy flux along the coast and averaged in a coastal fringe distant 1°
from the coast and 2° width. The maximum amplitude –reaching ~1 cm s$^{-1}$ µM– is concentrated in the
upper oxycline (Fig. 16a) with a peak during Austral winter. The peak season is also confirmed by the
EOF analysis of the climatological DO eddy flux (not shown). Despite the relative large meridional
variability in the amplitude, the mean vertical structure of the DO eddy flux consists in an approximate
exponentially decaying profile with depth, with a decay scale of ~90 m (Fig. 16b) so that at 300 m the
seasonal DO eddy flux is only 19% of that at 100 m on average along the coast. Figure 16a also reveals
that the annual DO eddy flux is larger towards the northern rim of the domain and extends deeper than
towards the south. The high values are increasingly confined close to the surface towards the southern
part of the domain, in comparison to the northern part, although the vertical attenuation displays a
similar scale.
## 4.3   Meridional boundaries
Here, our objective is to document the seasonality of the DO eddy flux. As a first step, we estimate the
distribution of mean DO eddy flux, in order to identify the regions where its magnitude is large and
thus where it is likely to vary seasonally with a significant amplitude.
**Mean seasonal flux**
The horizontal distribution of mean DO eddy flux displays the highest values at the boundaries of the
OMZ core (Fig. 17), and adjacent to the 45 µM isopleth. Towards the inner OMZ, the mean DO eddy
flux values decrease notoriously, with a factor of nearly 10 between the interior and exterior of the 10
µM contour. In agreement with the observations reported in the previous section, the mean DO eddy
flux decreases sharply with depth (approximately one order of magnitude between 100 m and 700 m),
with the highest values concentrated near the oxycline as expected from the increasing oxygen
concentration in this part of the OMZ. In this sense, the pattern of DO eddy flux around the depth of
the oxycline encloses a region of high variability (not shown).
To gain further insight with respect to the vertical structure of the DO eddy flux and at the same time,
diagnose the role of the mesoscale activity at the boundaries of the OMZ, we compute the mean DO

eddy flux across the two sections that correspond to the northern and southern limits of the OMZ (depicted in Fig. 18). These limits are defined based on Figure 17, and are located in the provinces of high amplitude of the mean DO eddy flux.

The DO eddy flux across each of the north and south boundaries was computed by averaging the product of the fluctuating velocity component normal to the boundary in the horizontal directions and the fluctuating DO concentration component, thereby obtaining horizontal eddy fluxes.

As observed in Figure 17, the highest values for both north and south boundary sections are also comprised between the oxycline and the lower OMZ core limit (Fig. 18), being almost one order of magnitude smaller at greater depths (Fig. 18c). These high values, located between ~100-300 m, are followed by a sharp decrease (average decrease of 1.5 cm s$^{-1}$ µM in 100 m). At the range of depths between 100 m and 300 m, the DO eddy flux displays higher values at the southern boundary (nearly twice as large) when compared with the northern boundary. This relationship is less clear when analyzing the lower part of the OMZ. At both meridional boundaries, the mean DO eddy flux in the upper part of the OMZ is nearly one order of magnitude larger than in the lower part.

**Seasonal eddy flux**

We now document the seasonal variability of the DO eddy flux across the OMZ boundaries analyzed above (Fig. 18). An EOF analysis of the mean seasonal cycle of the DO eddy flux is performed at the boundary sections previously defined. The Figure 19 presents the first EOF mode patterns along with the associated timeseries. In order to estimate the uncertainty associated with the location of the OMZ boundaries, we repeated this analysis for 12 nearby sections parallel to the boundaries and spaced by ~20Km. This leads to an estimated error (standard deviation across the different sections) of the DO eddy flux. The error is represented as a colored shading in the Figures 19bde. At both locations, the first EOF accounts for a well defined seasonal cycle. At the northern boundary (Fig. 19a), the seasonal cycle of the DO eddy flux peaks in Austral Winter, in phase with the DO changes along the coast (Fig. 16). Note that the seasonal cycle is in phase with the one of the intraseasonal activity of the horizontal current normal to the section, which was estimated the same way than the climatological eddy flux (see red line in Fig. 19b), supporting the idea that the climatological DO eddy flux results from anomalous advection. The amplitude of the mode pattern is maximum at the oxycline with DO between 20 and 45 µM, and presents a sharp decrease below the OMZ core depth (Fig. 19a). This sharp decrease is evidenced by the mean vertical profile of the DO eddy flux seasonal variability estimated as the RMS

across the section of the EOF mode pattern (Fig. 19e). The vertical structure of the DO eddy flux
variability indicates that there is a difference of nearly one order of magnitude between 100 and 300 m
depth. From that depth on, the DO eddy flux variability decreases linearly.
In contrast with the northern boundary, the seasonal variability at the southern boundary peaks during
Austral Spring (Fig. 19d), in phase with the intraseasonal activity of the horizontal currents normal to
the section. The amplitude of the seasonal cycle is the largest around the depth of the oxycline, and
remains high down to the vicinity of the OMZ core upper limit (Fig. 19c). Below the depth of the OMZ
core, the amplitude of the EOF mode decreases sharply (~one order of magnitude in 100m; Fig. 19c).
This is evidenced by the profile of the DO eddy flux seasonal variability, estimated in the same manner
as for the northern boundary (Fig. 19e). This profile shares some characteristics with its counterpart at
the northern boundary, meaning, a sharp decrease between the oxycline and the OMZ core depths,
suffering a reduction of nearly 90% (Fig. 19e). On the other hand, the variability along the southern
boundary is ~70% larger than along the northern boundary. At both boundaries, the zonal wavelength
of the seasonal DO eddy flux variability along the boundary is estimated to be in the order of ~$10^2$ km,
a scale that falls within the range of observed eddies diameter (Chaigneau and Pizarro, 2005), which
indicates that locally there can be an injection or removal of DO across the boundary on average over a
season. The mean DO eddy flux across the boundaries is nevertheless positive.

## 5 Discussion

We now discuss some limitations and implications of our results. While the model realistically
simulates the main characteristics of the OMZ (position, intensity, average volume and seasonal
variations), it still presents biases that could be influential on our results. In particular, and since the
coastal domain is viewed here as a boundary of the OMZ, it is important to have a realistic mean DO
concentration there. Compared to CARS, the simulated suboxic volume is however underestimated by
~6%, and 85% of this error can be attributed to the coastal domain (fringe of 3° from the coast). This
bias could be due to several factors. Montes et al. (2014) observed variations of the suboxic volume in
the order of 5%, when contrasting two simulations that used different oceanic open boundary
conditions, which indicates a sensibility of the simulated OMZ to the physical parameters and the
representation of the Equatorial Current system. This bias could also be partly due to coastal sediments
processes (DO demanding processes) that are not represented in our simulation. Using a similar

configuration to the one used in the present study on the Namibian OMZ, Gutknecht et al. (2013a) observed that the differences between the simulated OMZ volume and CARS increased towards the shelf, which could be related to the exclusion of the DO demand from the sediments in the model. On the other hand, in a study on the impact of sediment biogeochemistry upon the water column biogeochemical cycles in the northern end of the California Current system, Bianucci et al. (2012) argued that the sediment denitrification is balanced by a nitrification in the water column, obtaining similar bottom DO concentrations between their experiments even when disregarding the DO sediment demand. Nevertheless, the interaction between the sediments processes and the water column in terms of DO consumption in the mid latitude upwelling systems is still unclear. Regarding this point, the inclusion of a sediment module in the current model setting is planned for future work.

Besides other likely sources of biases related to an imperfect model setting (e.g. use of relatively low resolution atmospheric forcings near the coast, absence of air-sea coupling at mesoscale, absence of coupling with benthic oxygen demand or consideration of $N_2$ fixation), another inherent limitation of our study is related with the difficulty to validate some aspects of the eddy field, in particular its vertical structure. This might be overcome in the future as the Argo coverage increases (cf. TPOS2020).

With the limitations of our regional modeling approach in mind, it is worthwhile discussing some implications of our results. While previous studies have mostly focused on the role of the mean DO eddy flux in shaping the OMZ boundaries (Resplandy et al., 2012; Brandt et al., 2015; Bettencourt et al., 2015), we have documented here the seasonal variability in the DO eddy flux at the OMZ boundaries, including the "eastern boundary" formed by the coastal system. We infer that the seasonality of the DO eddy flux is controlled by different physical processes depending on the boundary. At the "eastern boundary", there is a constructive coupling between eddies resulting from the instability of the PUC peaking in Austral winter, and the enhanced DO along the coast resulting from an increased horizontal and vertical diffusion at the same season.

At the northern boundary of the OMZ, the DO eddy flux is also related to the strong EKE around 5°S that peaks in Austral winter. Despite the fact that the OMZ northern boundary is embedded in the equatorial wave guide, since the intraseasonal Kelvin wave activity tends to peak in Austral summer (Illig et al., 2014) it can be ruled out that the seasonal cycle in DO eddy flux is strongly linked to the intraseasonal long equatorial waves. The results of Echevin et al. (2011) also suggest that the enhanced

mesoscale activity observed near the northern OMZ boundary during winter would not be related to the equatorial Kelvin wave activity, but rather to local variations of the local current system (intrinsic or induced by the local wind stress). Whether or not the strong EKE found there results from the instability of the coastal current system or of the EUC and the South Equatorial Current (SEC), would need to be explored.

Regarding the southern boundary, it is interesting to note that the DO eddy flux peaks in Austral spring, three months later than at the northern boundary. A possible mechanism driving the local variability observed at the southern section is the generation of local baroclinic instability and vorticity input from wind stress curl as observed for the California system (Kelly et al, 1998). The southern section lies within the northeast rim of the Southeast Pacific Anticyclone, and the peak in the seasonal DO eddy flux coincides with the reported intensity peak of the seasonal cycle of the Anticyclone, towards the end of the year (Rahn et al., 2015; Ancapichún and Garcés-Vargas, 2015). In this sense, the mesoscale activity in this region could be directly modulated by the winds. Dewitte et al. (2008) also report that intraseasonal (internal) variability in currents can originate from the interactions between the annual extra tropical Rossby wave and the mean circulation in a medium resolution oceanic regional model over this region, a process also observed and documented from a high-resolution model over the North Pacific (Qiu et al., 2013). The actual source of the eddy activity in this region would also deserve further investigation.

Our study also reveals that the most prominent propagating features in DO inside the OMZ at annual frequency is below ~300 m, where the seasonal DO flux follows approximately the theoretical WKB ray paths of the annual ETRW. From that depth, the seasonal variability in physical fluxes becomes one order of magnitude larger than the one of the biogeochemical fluxes (Fig. 10c). This supports the observation that DO tends to behave as a passive tracer so that vertical displacements of the DO isopleths mimic those of the isotherms, inducing a seasonal DO flux that resembles the energy flux path of the ETRW. This mechanism adds a dimension to the understanding of the OMZ variability, considering that the vertical propagation of ETRW can take place at frequencies ranging from annual (Dewitte et al., 2008) to interannual (Ramos et al., 2008).

We now discuss some implications of our results with regards to current concerns around OMZ variability at long timescales. A recent study has suggested a trend in the OMZ towards expansion and intensification (Stramma et al., 2008) whose forcing mechanism remains unclear (Stramma et al.,

2010). Observations in the Pacific Ocean also suggest that the OMZ characteristics vary decadally
(Stramma et al., 2008; 2010). Since decadal variability can manifest as a low frequency modulation of
the seasonal cycle, our study may provide guidance for investigating OMZ variability at long
timescales. In particular we find that the amplitude of the seasonal cycle is nearly twice as large at the
southern boundary than at the northern boundary and "coastal boundary", which suggests a larger
sensitivity of the OMZ variability at the southern boundary to the modulation of eddy activity by
climate forcing. This view would preferentially link the OMZ low frequency fluctuations to mid
latitudes changes in the circulation. We note however that the relative contribution of the mean DO flux
and the DO eddy flux exhibits significant interannual fluctuations at the boundaries (not shown), which
suggests that eddy induced DO flux may not be the only key player for understanding long term trend
in the OMZ. It is interesting to note that so far, it has been difficult to reconcile the observed trend in
the OMZ with the trend simulated by the current generation of coupled models (Stramma et al., 2012),
which has been attributed to biases in the mean circulation and inadequate remineralization
representation (Cocco et al., 2013; Cabré et al., 2015). Our results support the view that such
discrepancy may partly originate from the inability of the low resolution models to account for the DO
eddy flux and its modulation. Regional modeling experiments also showed that eddy activity can be
modulated at ENSO and decadal timescales (Combes et al., 2015; Dewitte et al., 2012). This issue
would certainly require further investigation, and could benefit from the experimentation with our
coupled model platform. This is planned for future work.
Lastly, the seasonal changes in the OMZ evidenced in this work are associated with a seasonal change
of the oxycline depth (and an oxycline intensity change; not shown), which can be considered a proxy
for the production of greenhouse gases ($CO_2$ and $N_2O$) inside the OMZ (e.g. Paulmier et al., 2011;
Kock et al., 2016). Our results suggest that the impact of the OMZ on the atmosphere through the
production of climatically-active gases, such as $CO_2$ and $N_2O$, would be seasonally damped during
austral winter, due to a deepening of the oxycline and a weakening of its intensity.

## 6   Summary and conclusions

A high resolution coupled physical/biogeochemical model experiment is used to document the seasonal
variability of the OMZ off Peru. The annual harmonic of DO reveals three main regions with enhanced
amplitude or specific propagation characteristics, suggesting distinct dynamical regimes: 1) The coastal
domain; 2) the offshore ocean below 400 m and 3) at the southern and northern boundaries. In the
coastal portion of the OMZ, the seasonal variability is related to the local wind forcing, and therefore
follows to a large extent the paradigm of upwelling triggered productivity, followed by
remineralization. It is shown in particular that the DO peaks in Austral winter which is associated with
horizontal and vertical diffusion induced by both the increase in baroclinic instability and intraseasonal
wind activity. This is counter intuitive with regards to the seasonality of the alongshore upwelling
favorable winds also peaking in Austral winter, which would tend to favor the intrusion of
deoxygenated waters from the open ocean OMZ to the shelf. Instead, the coastal domain can be viewed
as a source of DO in Austral winter for the OMZ through offshore transport. The latter is induced by
eddies that are triggered by the instabilities of the PUC. In the model, the offshore DO eddy flux has a
marked seasonal cycle that is in phase with the seasonal cycle of the DO along the coast, implying that
the coastal domain, viewed here as the eastern boundary of the OMZ, is a source of seasonal variability
for the OMZ. This appears to operate effectively in the upper 300 m. Below that depth, the DO eddy
flux is much reduced due to both a much weaker eddy activity, and very low DO concentration. On the
other hand, a mean seasonal DO flux is observed and exhibits propagating features reminiscent of the
vertical propagation of energy associated with the annual extra tropical Rossby wave.
In the upper 300 m, the OMZ seasonal variability is also associated with the DO eddy flux at the OMZ
meridional boundaries where it is the most intense. We find that the seasonal cycle in DO eddy flux
peaks in Austral winter at the northern boundary, while it peaks a season later at the southern boundary.
Additionally, the amplitude of the seasonal cycle in DO eddy flux is larger at the southern boundary
than at the northern boundary. The schematic of Figure 20 summarizes the main processes documented
in this paper to explain the seasonality of the OMZ.
**Acknowledgments**
O. Vergara was supported by a doctoral scholarship from the National Chilean Research and
Technology Council (CONICYT) through the program Becas Chile (scholarship 72130138). The
authors are thankful for the financial support received from the Centre National d'Etudes Spatiales
(CNES). During the preparation of this work O. Vergara was supported by a mobility scholarship from
the University of Toulouse, through the ATUPS program when visiting the CEAZA. M. Ramos
acknowledges support from FONDECYT (project 1140845) and Chilean Millennium Initiative

1  (NC120030). O. Pizarro acknowledges support from the FONDECYT 1121041 project and the Chilean

2  Millennium Initiative (IC-120019). The authors thank the two anonymous reviewers for their

3  constructive comments that helped improving the manuscript.

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

**Table 1.** Austral summer (DJF mean) and winter (JJA mean) seasonal anomalies of the DO budget,
averaged over the core of the Peru Under Current at 12°S (as depicted by the red contour in Figure 12).
The values for the seasonal cycle and the reconstructed first EOF mode (Figures 12 and 13) are
presented along with the difference Climatology-EOF. All values are in $10^{-6}\mu$M s$^{-1}$. Mixing here
consists in the summep-up contribution of horizontal diffusion and ( $K_h \nabla^2 O_2$ ) and vertical diffusivity (
$\frac{\partial}{\partial z}\left(K_z \frac{\partial O_2}{\partial z}\right)$ ).

|  | Climatology | | EOF | | Difference | |
| --- | --- | --- | --- | --- | --- | --- |
|  | **Summer** | **Winter** | **Summer** | **Winter** | **Summer** | **Winter** |
| **dO$_2$/dt** | 1.10 | -2.74 | 1.30 | -2.67 | -0.2 | -0.07 |
| **Adv** | 0.61 | -9.38 | 0.85 | -9.30 | -0.24 | -0.08 |
| **Mixing** | -0.42 | 7.99 | -0.35 | 7.99 | -0.07 | 0.0 |
| **Biogeochemical Flux** | 0.91 | -1.35 | 1.00 | -1.35 | -0.09 | 0.0 |

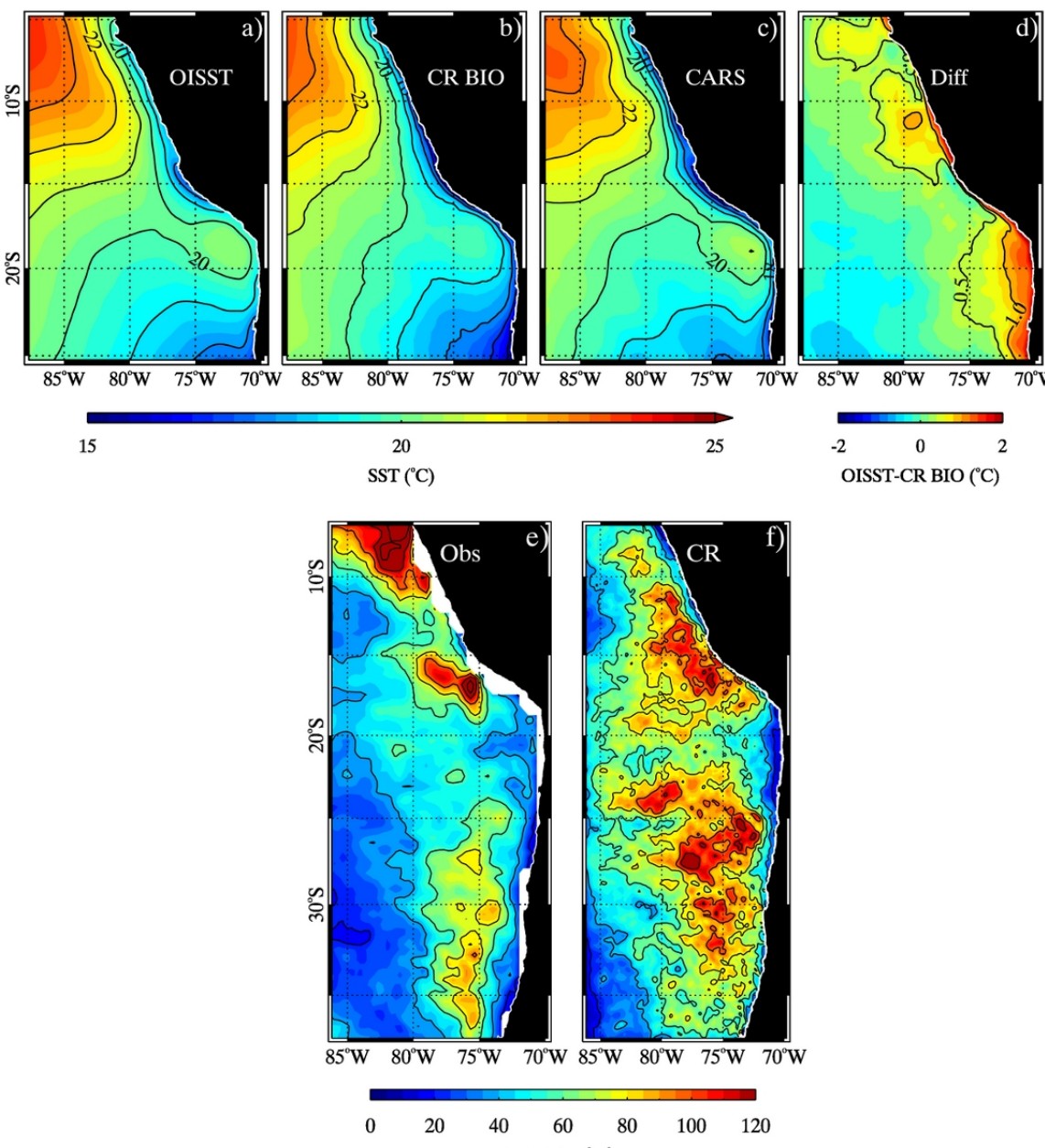

**Figure 1.** Mean sea surface temperature (SST) between 2000 and 2008 for (a) OISST product (0.25°x0.25°), (b) the simulation (1°/12) and (c) CARS dataset (0.5°x0.5°). (d) Difference between the OISST product and the simulation. Mean Eddy Kinetic Energy (EKE) between 1993 and 2008, for (e) TOPEX/Poseidon Jason 1-2 merged product (0.25°x0.25°), and (f) Simulation (1°/12). EKE was derived from the interannual anomalies of the geostrophic velocity field.

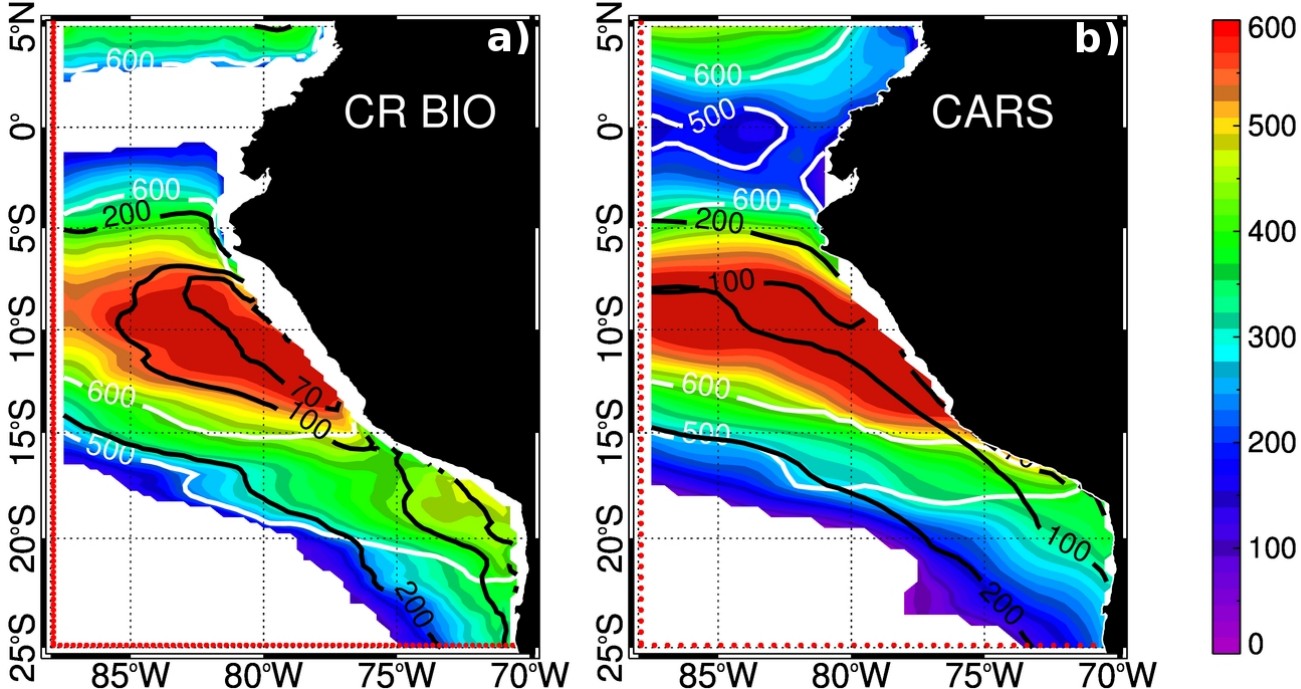

Figure 2. Mean Oxygen Minimum Zone core thickness (color scale in meters) for (a) the simulation and (b) CARS. Depth of the lower (white) and upper (black) limits of the OMZ core are also depicted. The OMZ core is defined as [DO] < 20 µM. The red dots denote the horizontal resolution of the DO field.

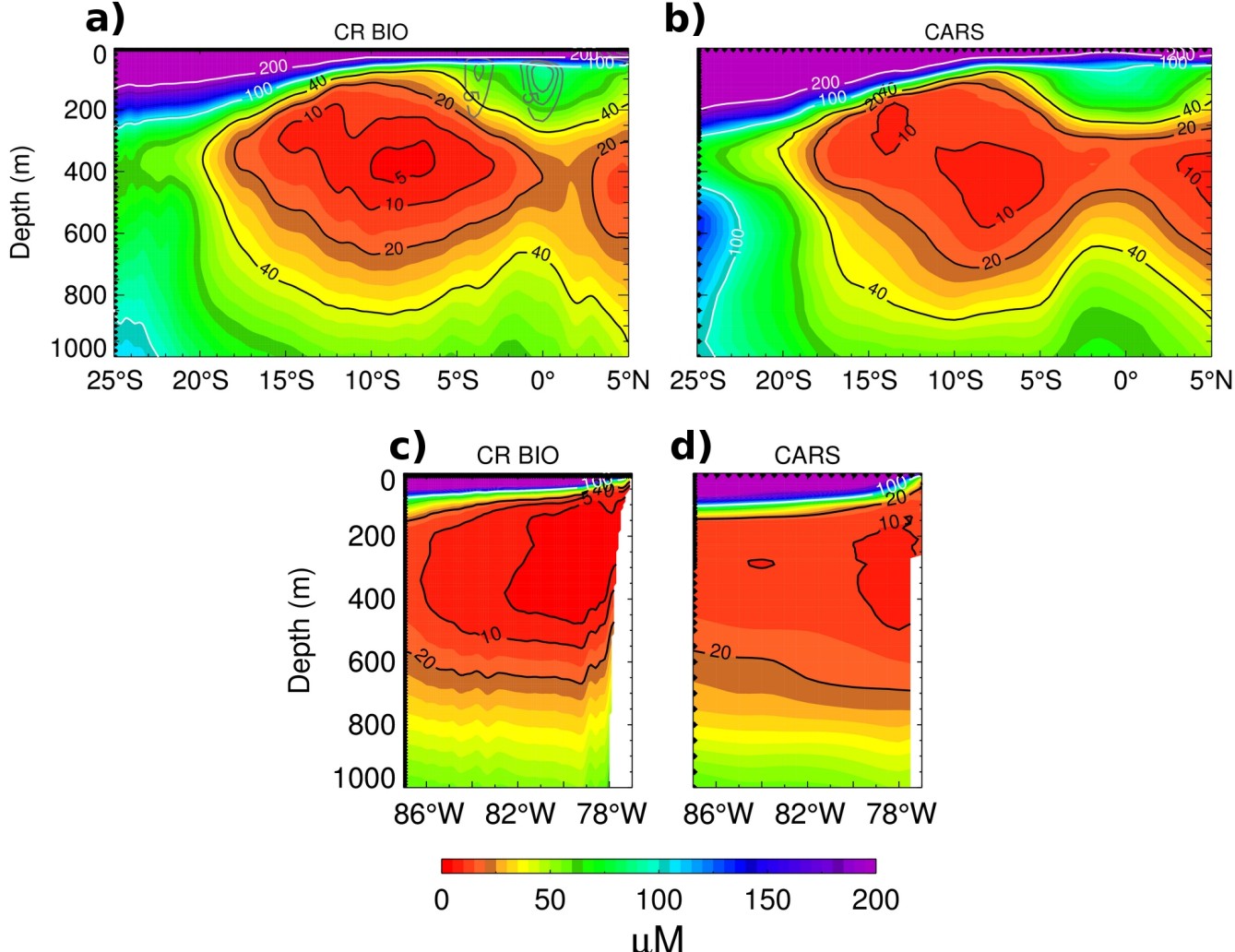

**Figure 3.** Mean oxygen concentration for a meridional section at 85°W (a and b) and a cross shore section at 12°S (c and d), for both the simulation and CARS. Gray contours in (a) show mean zonal speed of 5,10 and 15 cm s$^{-1}$ respectively. The black dots denote the horizontal and vertical resolution of the DO field.

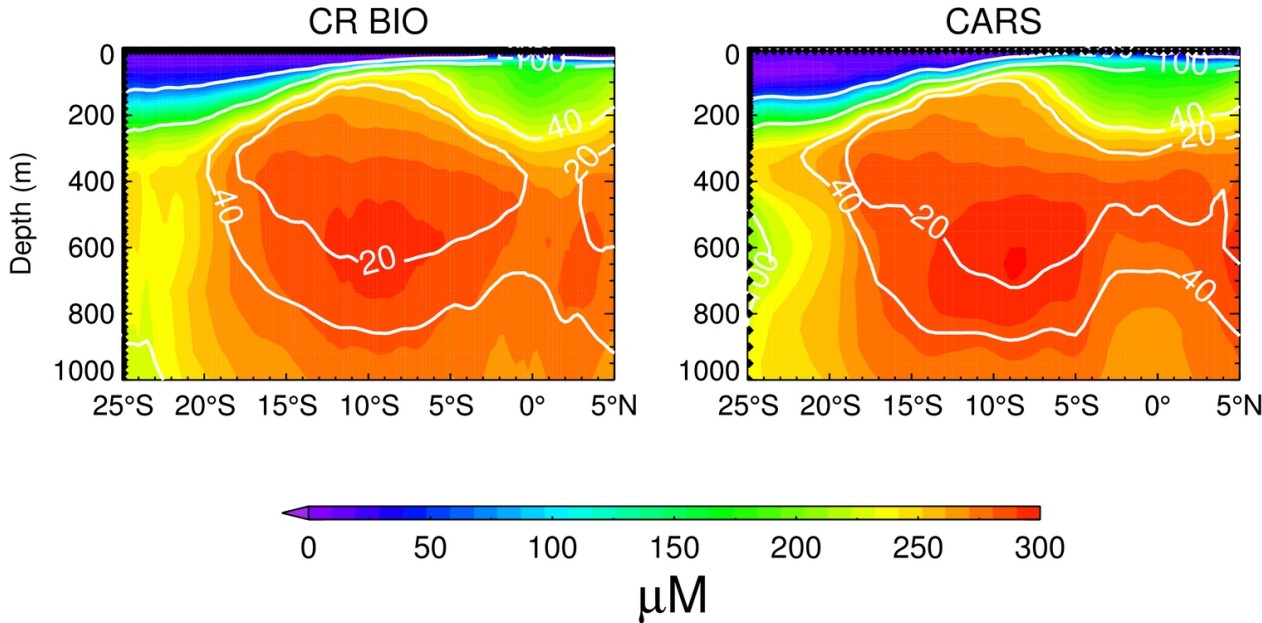

Figure 4. Mean Apparent Oxygen Utilization (AOU) at 85°W for both CR BIO and CARS. White contours denote the mean oxygen concentration isopleths (in µM). The black dots denote the horizontal and vertical resolution of the DO field.

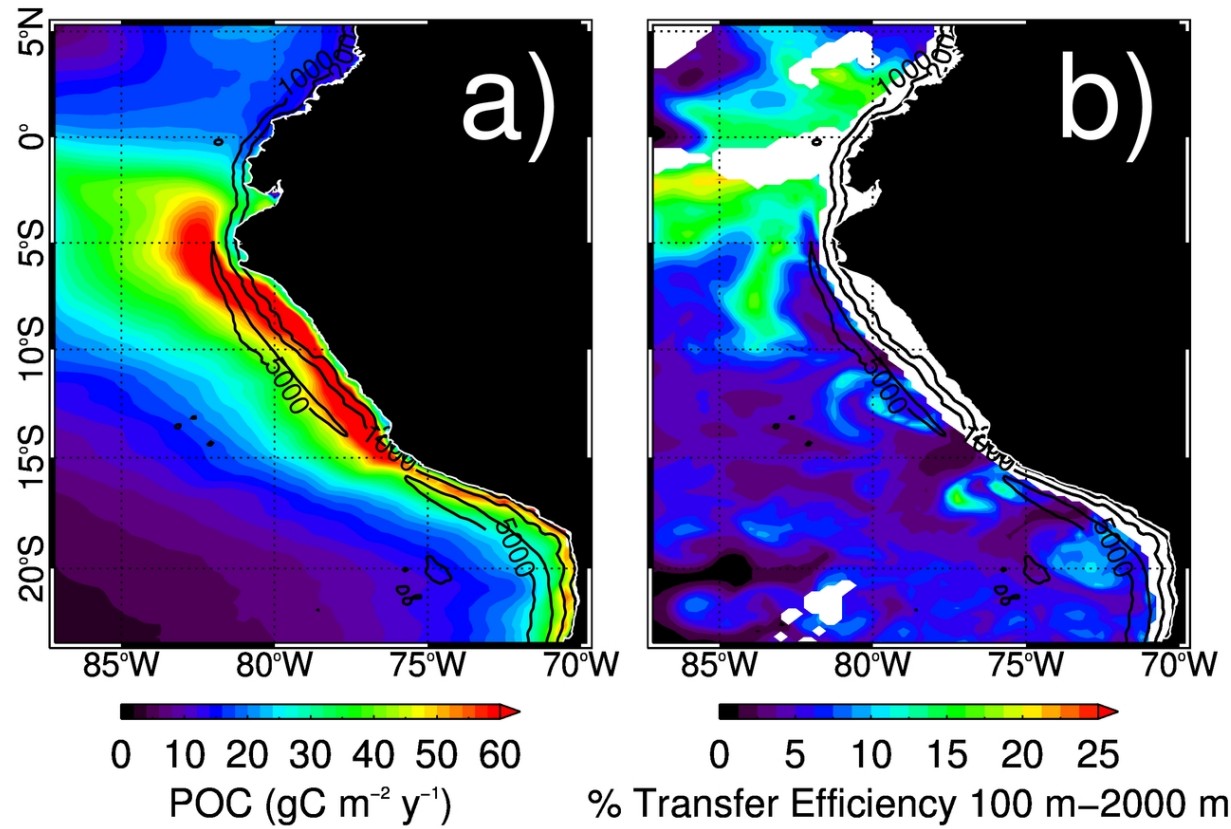

**Figure 5.** (a) Particulate Organic Carbon (POC) flux at 100 m and (b) POC transfer efficiency between 100 m and 2000 m (POC flux at 2000 m divided by POC flux at 100 m), computed from the simulation. Integrated carbon flux at the depth of 100 m: 0.8 Pg C year$^{-1}$. Black contours correspond to the 200, 1000 and 5000 meters isobaths.

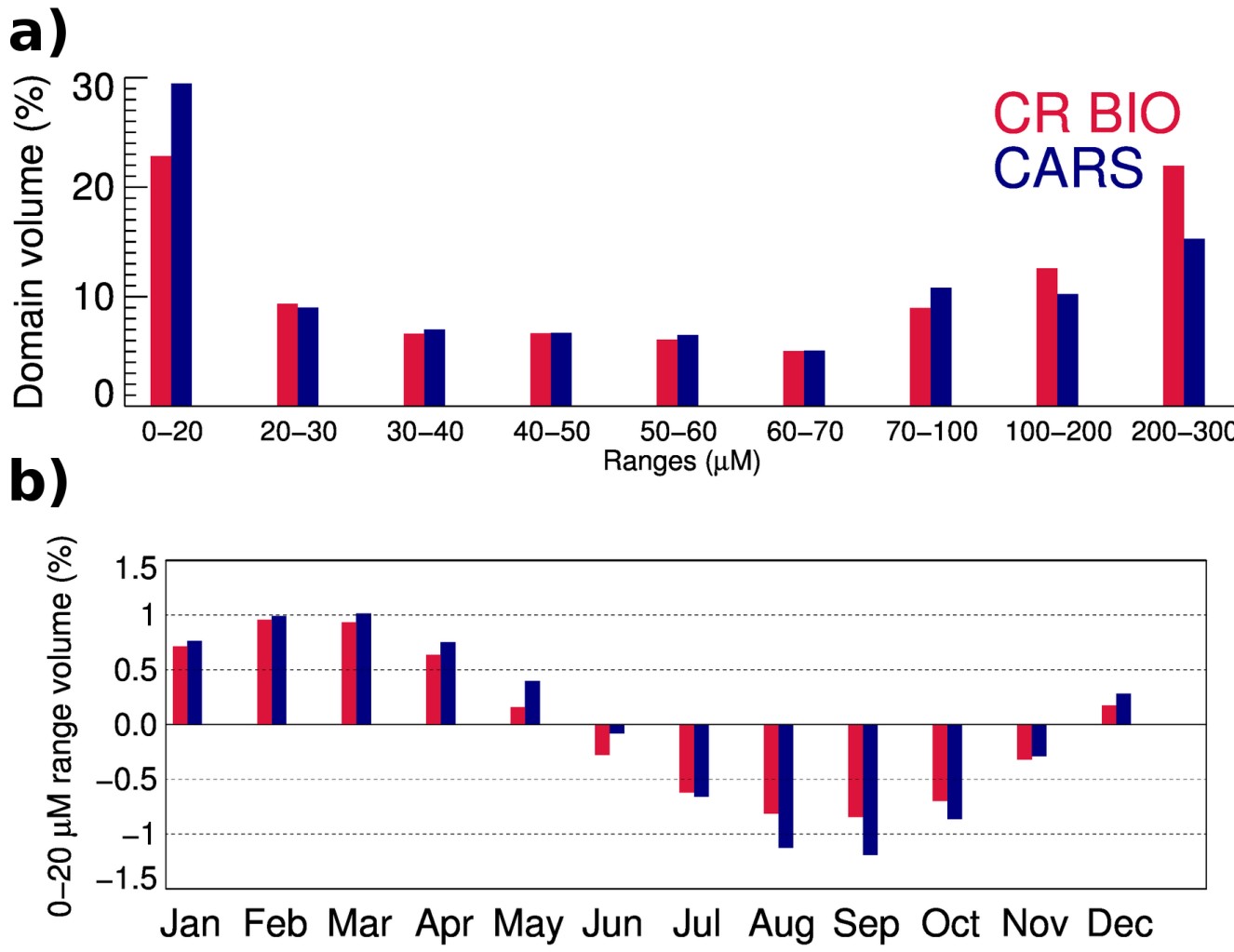

**Figure 6.** (a) Domain volume distribution (25°S-5°N, 88°W-70°W) as a function of the oxygen concentration, and (b) annual cycle, relative to the mean, of the volume distribution inside the OMZ core (DO value range corresponding to 0-20 µmol L$^{-1}$), for both CARS and the simulation.

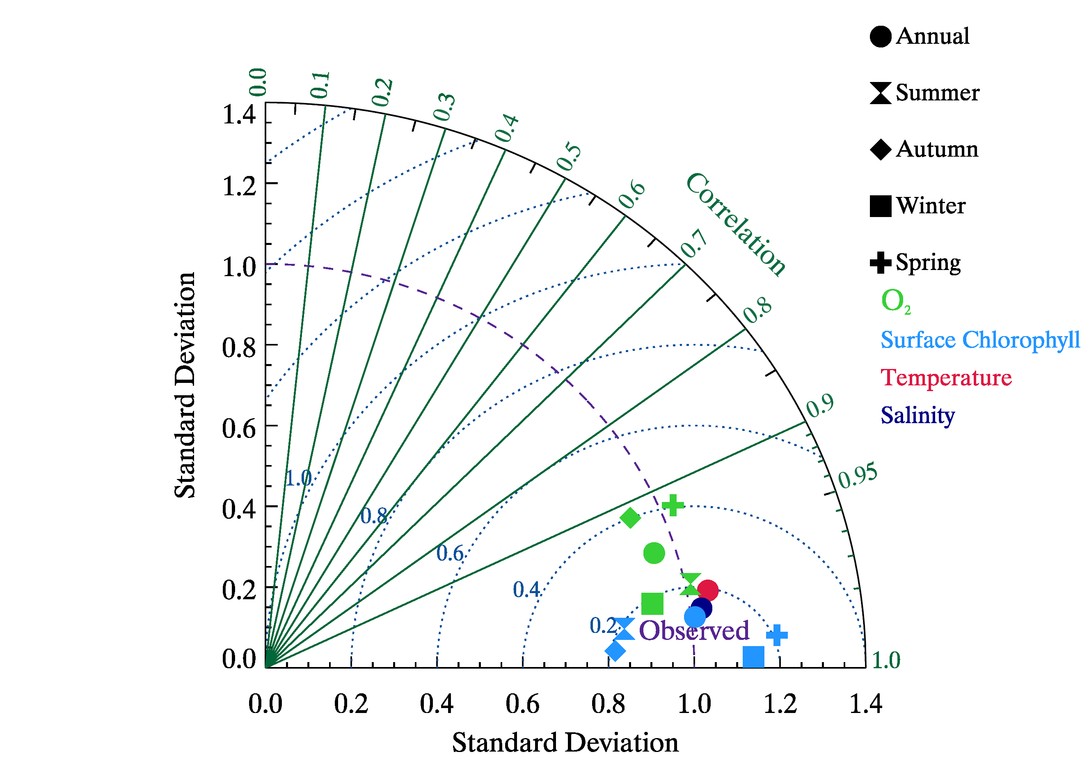

Figure 7. Taylor diagram of the seasonal mean (hourglass, diamond, square and cross) and annual mean (circle) pattern of DO and Surface Chlorophyll (25°S-5°N, 88°W-70°W). Only annual mean pattern comparisons are shown for temperature and salinity (same spatial domain). DO, temperature and salinity were vertically averaged between 100 and 600m depth (focus on the OMZ core). Only the surface chlorophyll values within 250 km next to the coast were considered. The comparisons are made between the simulation and CARS (for DO, temperature and salinity) and SeaWiFS (for surface chlorophyll). Ordinate and abscissa axes represent the standard deviation normalized by the observations standard deviation. Blue dotted radial lines indicate the RMS difference between the observations and the simulation.

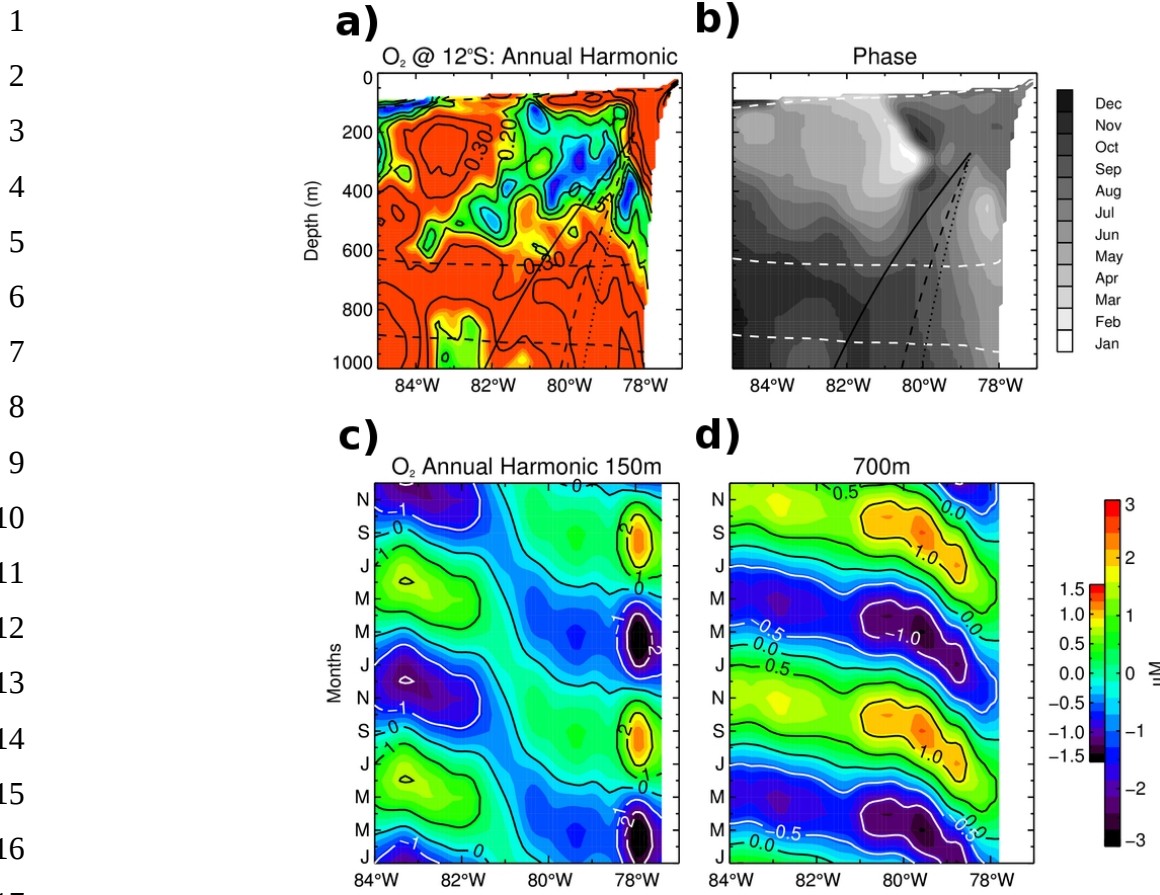

**Figure 8.** (a) Amplitude and (b) phase of the annual maximum (in months) of the annual harmonic of the normalized DO concentration at 12°S. The slanted vertical lines indicate the theoretical WKB ray paths at a frequency of $\omega = 2\pi \cdot 1 \text{year}^{-1}$, for different value of phase speed. The theoretical trajectories were computed using the phase speed of the first (full), second (dashed) and third (dotted) baroclinic modes of a long Rossby wave. Dashed contours in (a) and (b) depict the 45 and 20 µM mean DO values. Land and the region outside the 45 µM mean DO isopleth are masked in white. (c) Annual harmonic of the DO concentration at 12°S, at 150 m and (d) 700 m depth. Small color scale corresponds to 700 m and the large color scale denotes the levels used in (c).

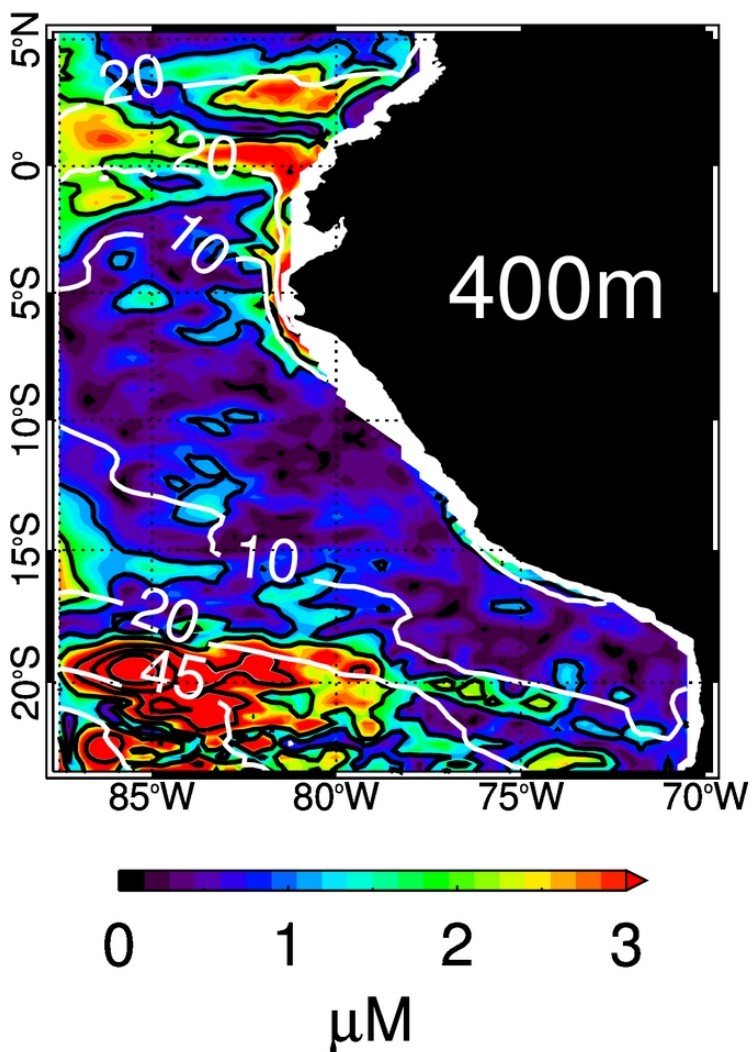

3   **Figure 9.** Annual DO harmonic amplitude at 400 meters depth. White contours denote the 10, 20 and

4   45 μM mean oxygen isolines. Black contours denote the 1, 2, 4, 6 and 8 μM levels.

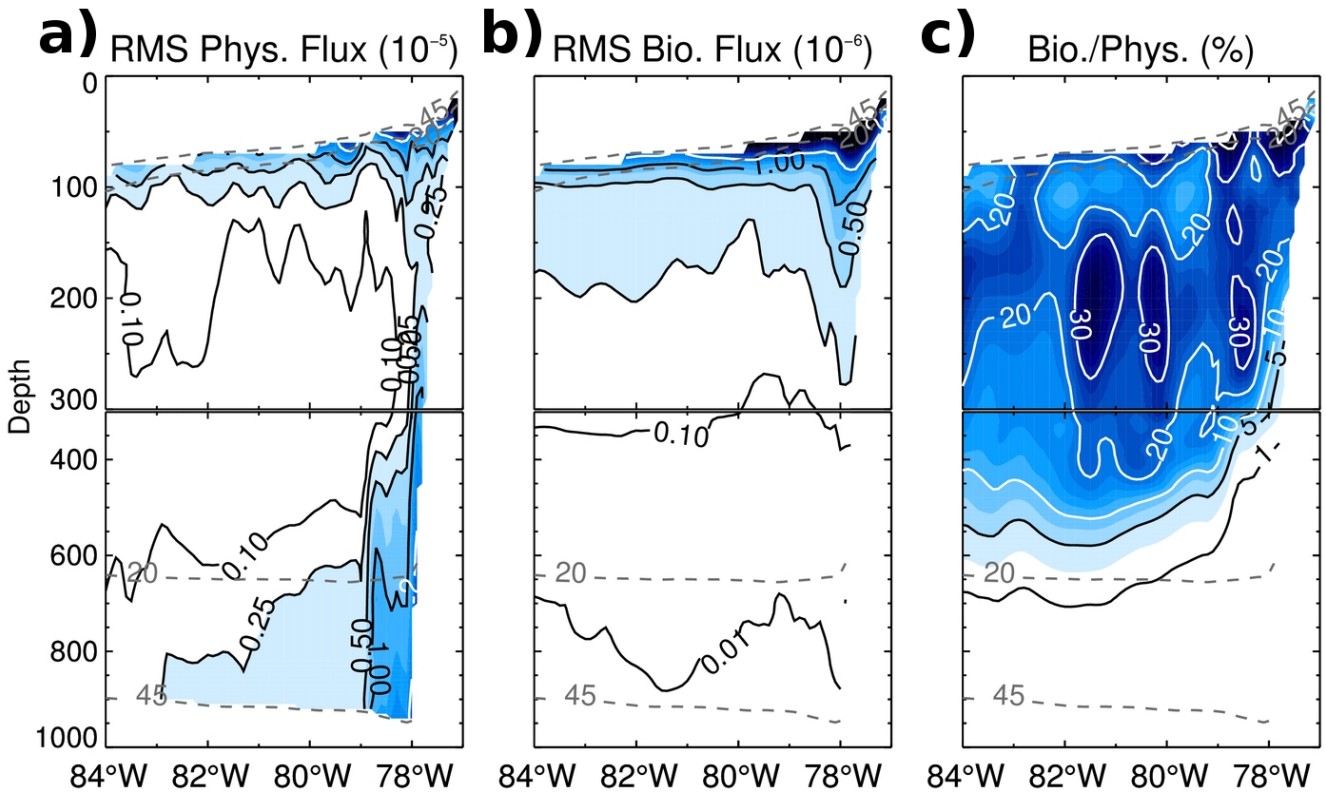

**Figure 10.** Root mean square of the seasonal cycle of (a) Physical and (b) Biogeochemical oxygen fluxes (in $10^{-5}$ μM s$^{-1}$ and $10^{-6}$ μM s$^{-1}$, respectively) for CR BIO at 12°S. c) Ratio between the RMS of the biogeochemical fluxes and the physical fluxes, expressed as percentage. Dashed contours depict the 45 and 20 μM mean oxygen values. Note the vertical scale change at 300m depth. Land and the region outside the 45 μM mean DO isopleth are masked in white.

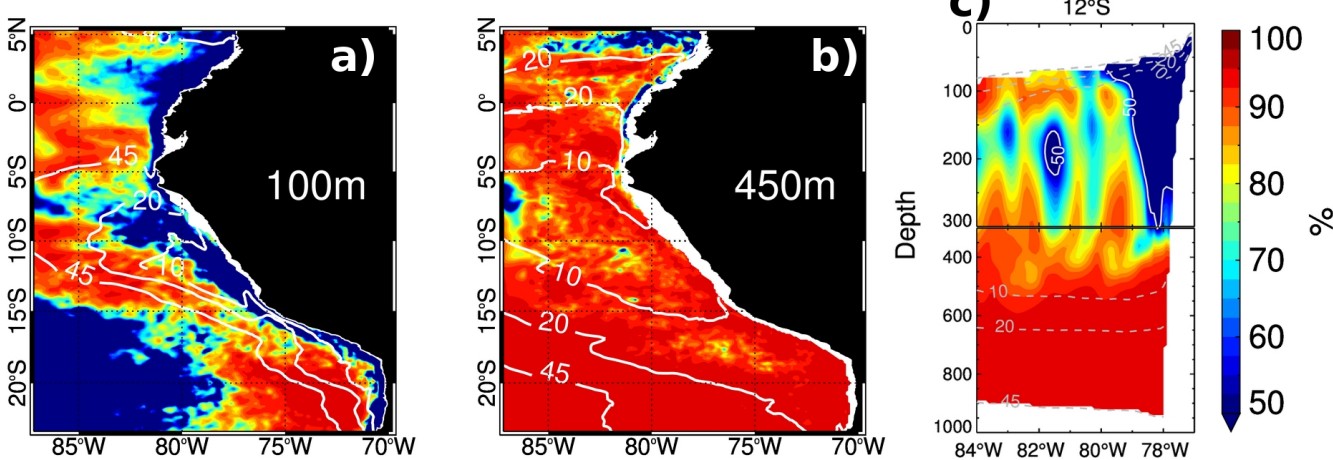

**Figure 11.** Percentage of the seasonal DO rate variance explained by the physical fluxes, at (a) 100 and (b) 450 meters depth, and along a cross shore section at 12°S. Solid white lines (a,b) and dashed gray lines (c) denote the 10, 20 and 45 µM mean DO isopleths. Land and the region outside the 45 µM mean DO isopleth are masked in white in (c).

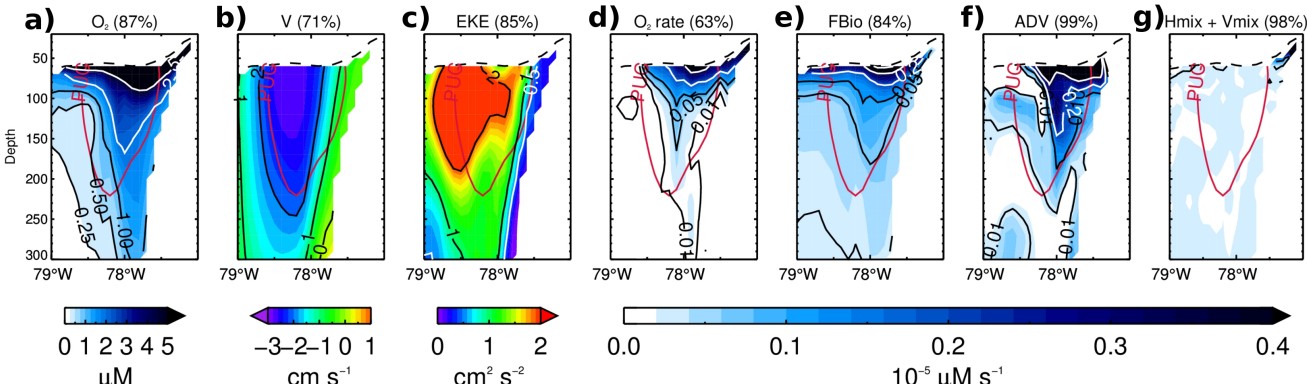

**Figure 12.** First EOF mode pattern of (a) DO, (b) alongshore currents component, (c) Eddy Kinetic Energy, (d) oxygen rate, (e) biogeochemical flux, (f) advective terms (sum of horizontal and vertical components) and (g) mixing terms (sum of horizontal and vertical components). Percentage of explained variance by each EOF mode pattern is indicated in parentheses on top of each panel. The red contour denotes the mean position of the Peru Under Current core, defined here as alongshore southward current exceeding 4 cm s$^{-1}$. The black dashed contour denotes the mean DO 45µM isopleth. Land and the region outside the 45 µM mean DO isopleth are masked in white. The EOF mode patterns were multiplied by the RMS of the PC timeseries. Multiplying the EOF pattern by the PC timeseries plotted in Figure 13 yields the contribution of the first EOF mode to the original field, in dimensionalized units (i.e. µM s$^{-1}$ for the tendency terms).

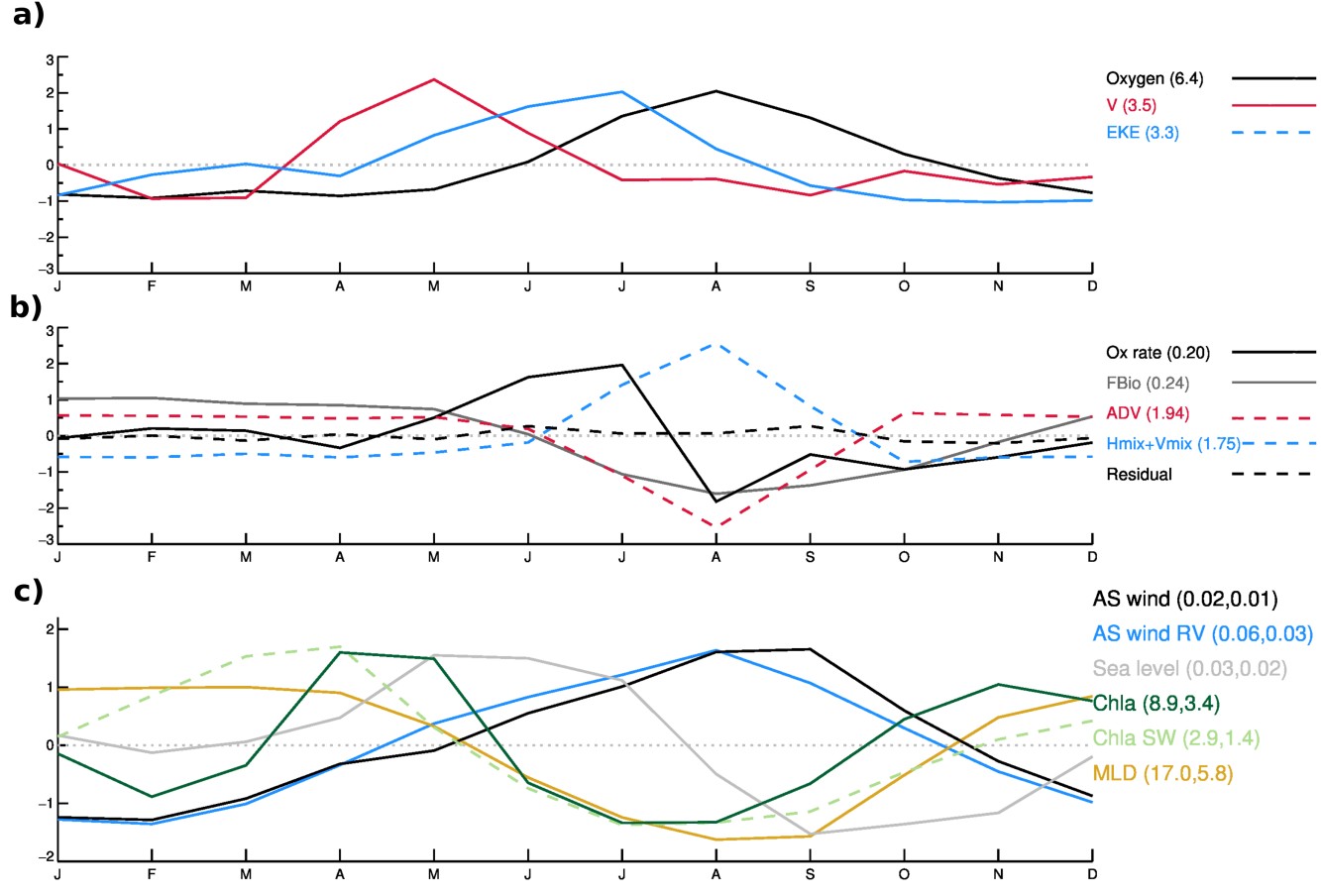

**Figure 13.** (a, b) Non dimensional principal components (PC) associated with the EOF patterns in Figure 12. Multiplying the principal component by the associated EOF pattern (from Fig. 12) yields a first EOF-mode reconstruction of the original field. RMS values of the principal components are indicated in parenthesis (corresponding units as in Fig. 12). The residual corresponds to the difference between the rate of DO change and the sum of all the terms of the rhs of Eq. 1 in terms of the normalized PC timeseries. The weak residual indicates that the seasonal DO budget can be interpreted from the EOF decomposition. The EOF decomposition was performed over the climatological (mean seasonal cycle) fields. (c) Normalized seasonal cycle of: coastal alongshore wind (AS wind) and coastal alongshore wind Running Variance (variance over a 30 day running window) at 12°S, sea level at the coast at 12°S, surface chlorophyll-a from CR BIO (Chla) and from SeaWiFS (Chla SW) averaged over a coastal band of 2° width at 12°S, and Mixed Layer Depth at the coast (MLD) at 12°S. Mean and RMS used to normalize each time series, are indicated in parenthesis. Original seasonal cycle is found by multiplying the normalized series by its RMS and then adding the mean. Original units are N m$^{-2}$, m, mg m$^{-3}$, and m respectively.

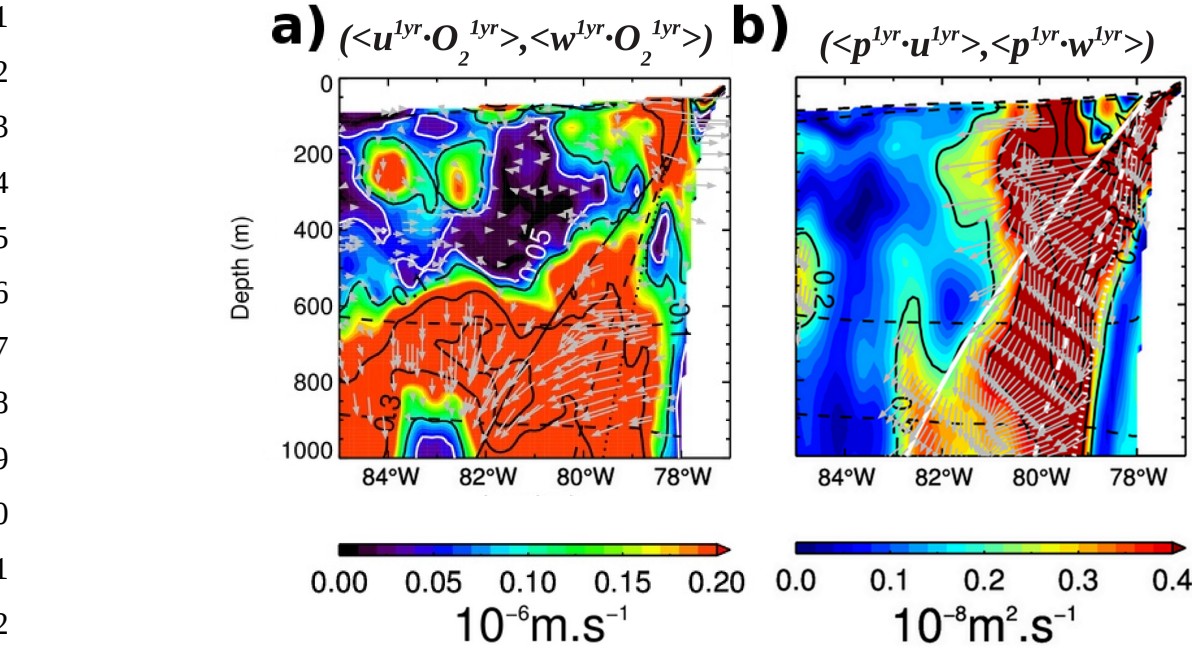

**a)** $(<u^{1yr}\cdot O_2^{\,1yr}>,<w^{1yr}\cdot O_2^{\,1yr}>)$  **b)** $(<p^{1yr}\cdot u^{1yr}>,<p^{1yr}\cdot w^{1yr}>)$

$10^{-6} \mathrm{m.s^{-1}}$

$10^{-8} \mathrm{m^2.s^{-1}}$

**Figure 14.** (a) Norm of the annual DO flux vector (i.e. $\sqrt{\left(\langle u^{1yr}\cdot O_2^{1yr}\rangle\right)^2+\left(\langle w^{1yr}\cdot O_2^{1yr}\rangle\right)^2}$ ) for a cross shore section at 12°S. Arrows indicate the vector direction (i.e. $\left[\langle u^{1yr}\cdot O_2^{1yr}\rangle,\langle w^{1yr}\cdot O_2^{1yr}\rangle\right]$ ). The DO signal was normalized by its Root Mean Square value before computing the annual harmonic, in order to emphasize the flux patterns where DO concentration is very low. (b) Norm of the annual energy flux vector (i.e. $\sqrt{\left(\langle p^{1yr}\cdot u^{1yr}\rangle\right)^2+\left(\langle p^{1yr}\cdot w^{1yr}\rangle\right)^2}$ ). Arrows inside the 0.2 contour indicate the vector direction (i.e. $\left[\langle p^{1yr}\cdot u^{1yr}\rangle,\langle p^{1yr}\cdot w^{1yr}\rangle\right]$ ). A range of theoretical WKB trajectories (1 year period) originating from near the coast at the surface are drawn for phase speed values of a first (full), second (dashed) and third (dotted) baroclinic modes. The range of phase speed values (modes 1-3) are obtained from a vertical mode decomposition of the mean model stratification. Dashed black contours indicate the 45 and 20 µM mean DO isopleths. Land and the region outside the 45 µM mean DO isopleth are masked in white.

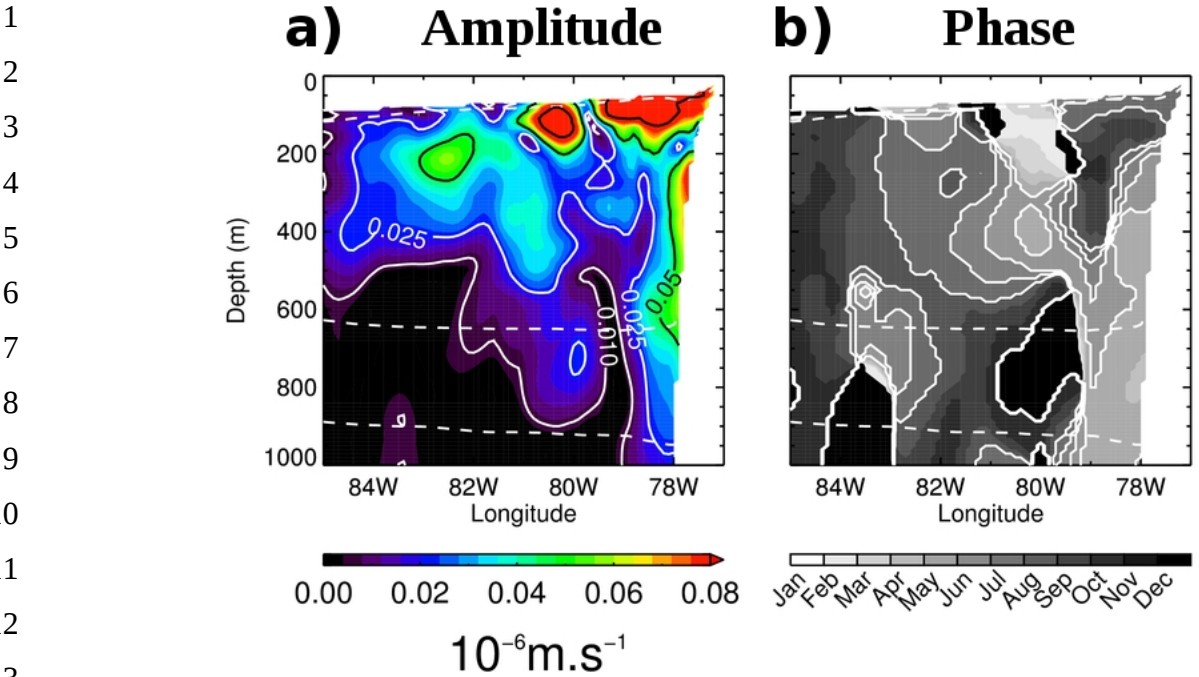

**Figure 15.** Zonal section of the annual harmonic of the module of the seasonal DO eddy flux vector $\left( \overline{\langle u' \cdot O_2' \rangle}, \overline{\langle w' \cdot O_2' \rangle} \right)$ at 12°S. (a) Amplitude of the harmonic and (b) phase of the annual maximum (in months). Dashed white contours indicate the 45 and 20 µM mean DO isopleths. DO was normalized by its RMS prior to carrying out analysis. Land and the region outside the 45 µM mean DO isopleth are masked in white.

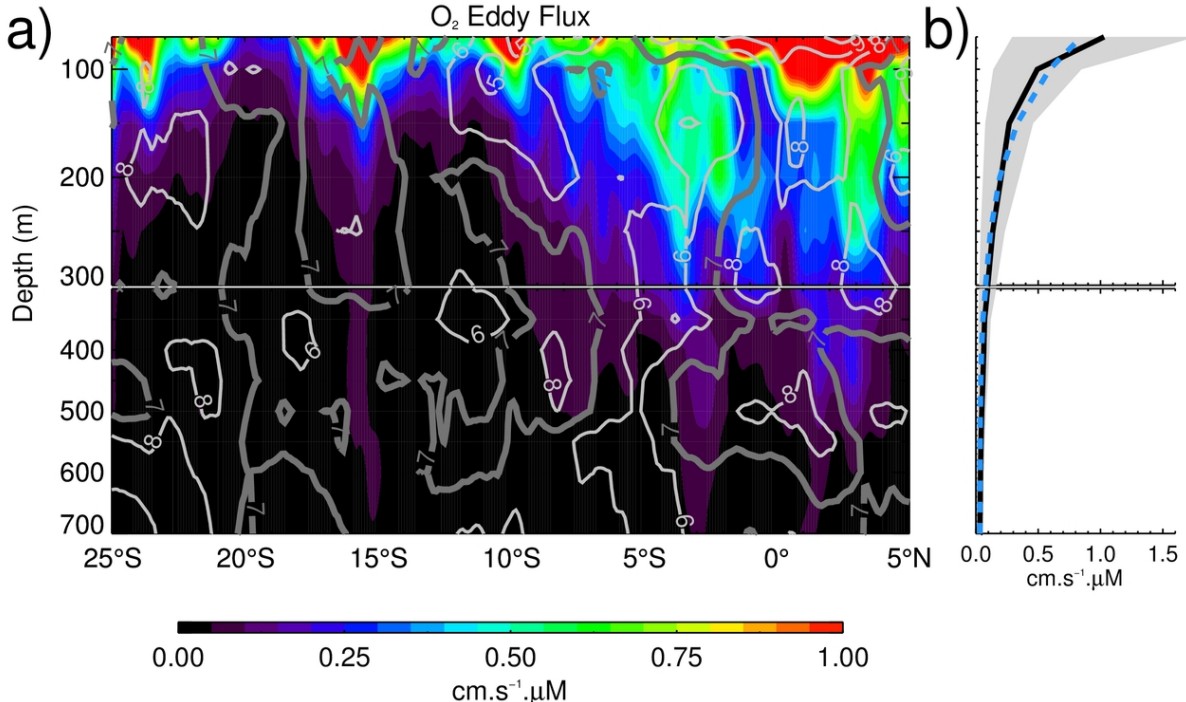

**Figure 16.** (a) Amplitude (color shading) and phase (months, gray contours) of the annual harmonic of
the climatological DO eddy flux along the coast. The climatology of DO eddy flux was averaged over a
coastal fringe of 2° width starting from 1° from the coast. (b) Meridional average vertical profile (black
line), +/- RMS (gray shading). An exponential model fitted to the average vertical profile (dashed blue
line) yields a vertical decay scale of ~90 m.

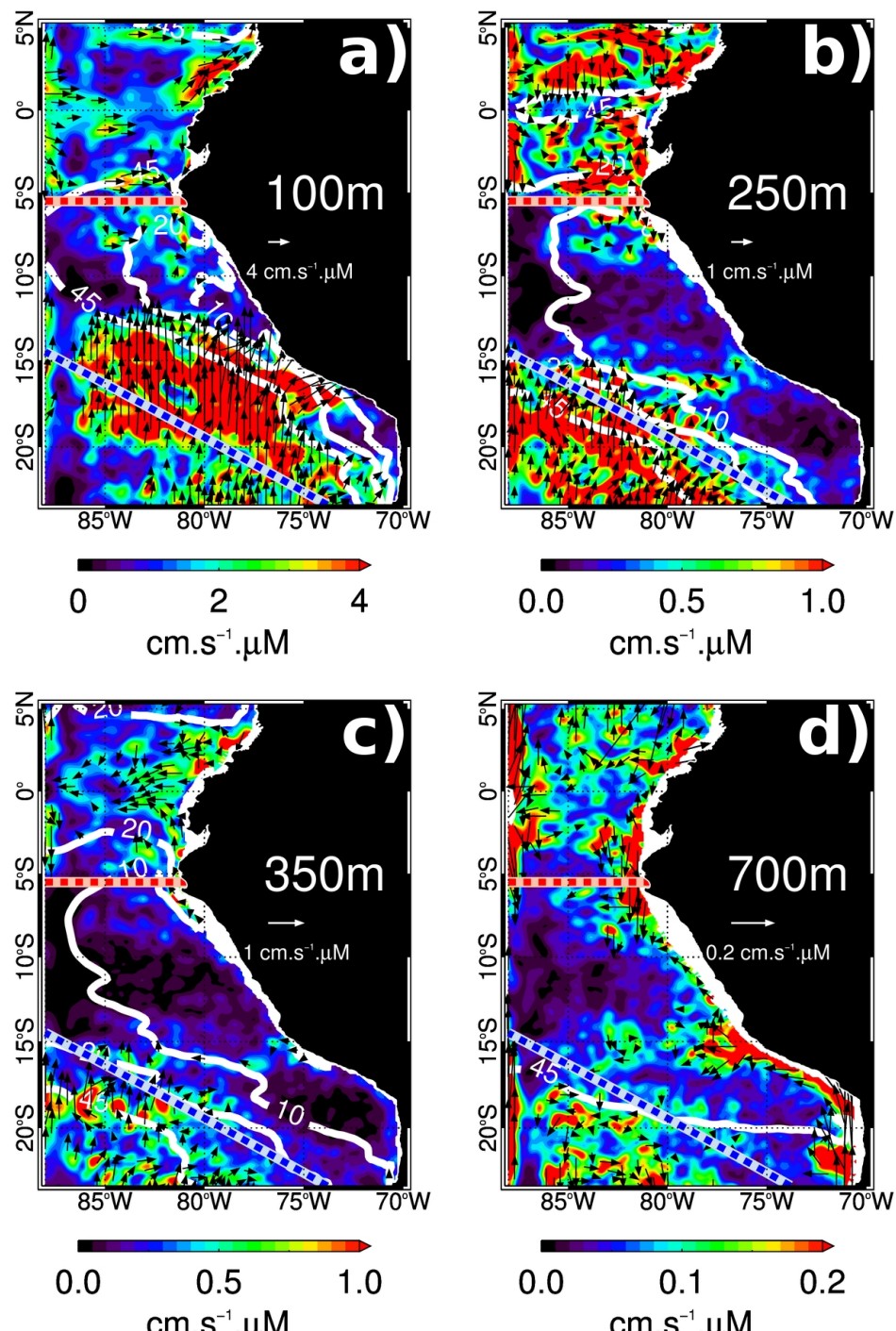

**Figure 17.** Module of the mean DO eddy flux vector $\left(\langle u' \cdot O_2' \rangle, \langle v' \cdot O_2' \rangle\right)$ at (a) 100 m, (b) 250 m, (c) 350 m and (d) 700 m depth. Arrows -displayed only for values above the central value in each colorbar- denote the vector direction and strength. White contours correspond to the 45, 20 and 10 µM mean DO values. Red and blue lines denote the position of vertical sections.

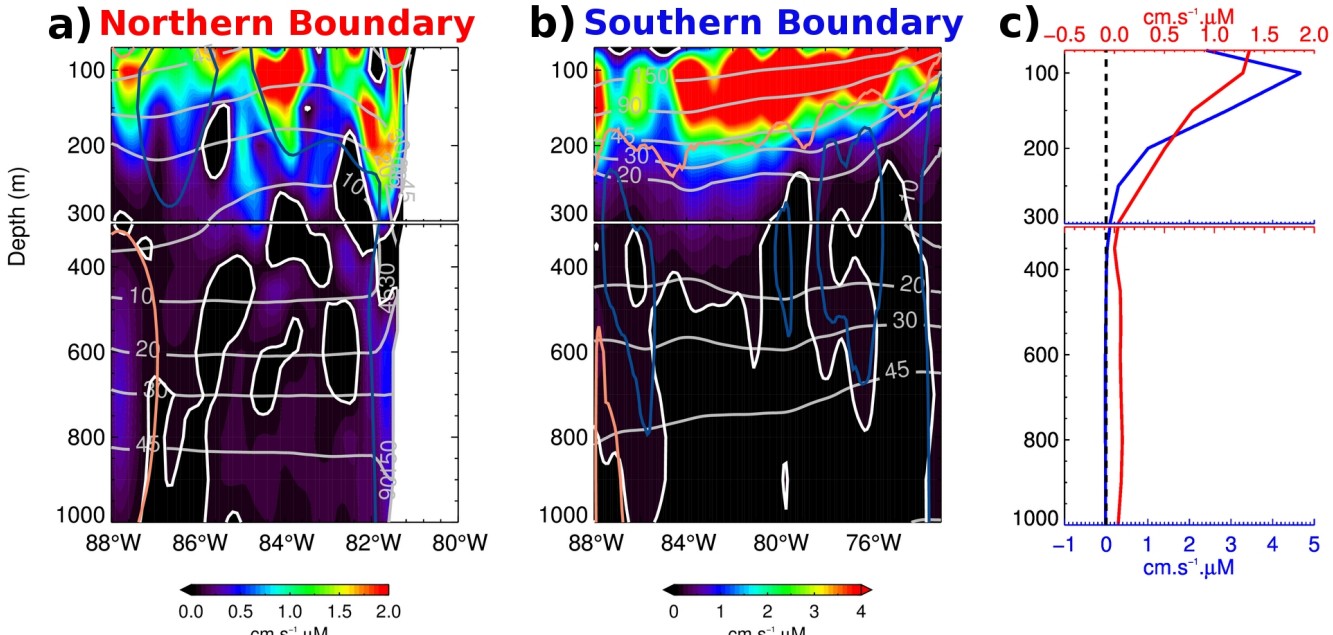

**Figure 18.** (a) Mean DO eddy flux normal to the section denoted by the red line in Fig. 17. (b) Mean DO eddy flux normal to the section denoted by the blue line in Figure 17. (c) Horizontal mean of (a) and (b) (red and blue lines, respectively). Gray contours denote mean DO concentrations, and light red/blue contours correspond to positive/negative values of mean currents normal to the section (1.0/-1.0 cm s$^{-1}$ in (a) and 0.4/-0.2 cm s$^{-1}$ in (b)). White contour denotes the 0 value. The sign convention was chosen so that a positive horizontal flux indicates transport towards the interior of the OMZ.

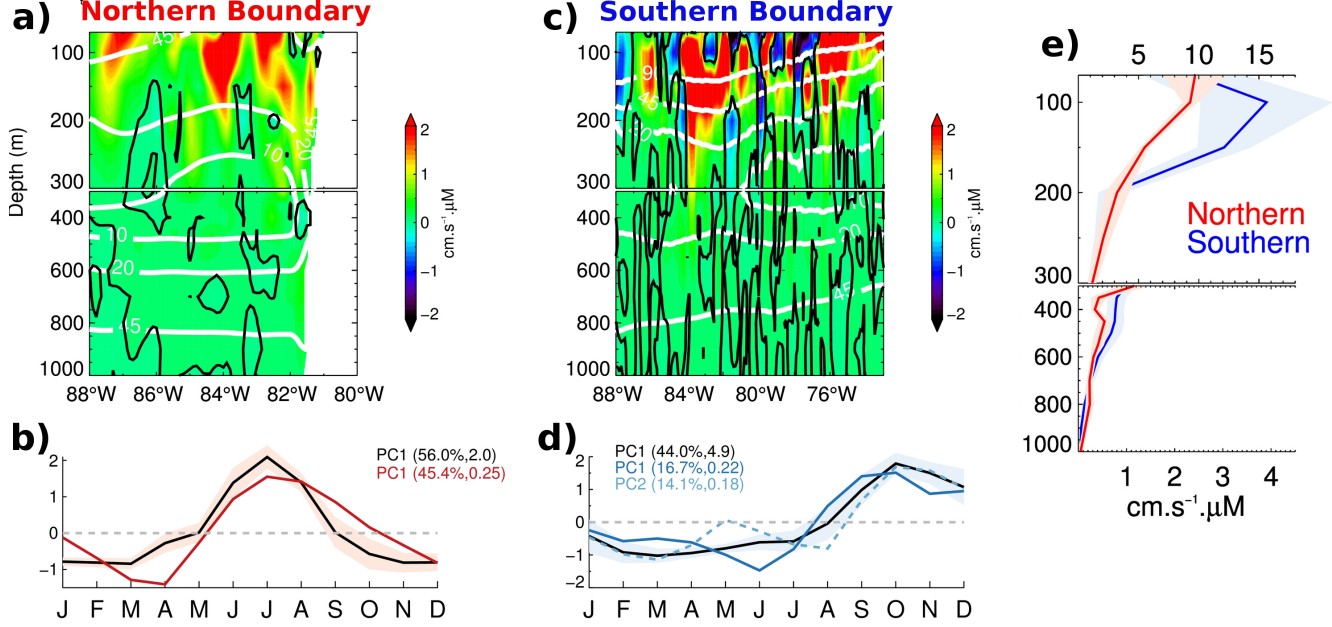

**Figure 19.** (a) First EOF mode of the seasonal cycle of the DO eddy flux normal to the section depicted in Fig. 17 by the dashed red line (Northern boundary). (b) Principal component (PC) timeseries associated with the first EOF mode (black line). The red line in (b) corresponds to the PC timeseries associated with the first EOF mode of the seasonal cycle of the 30-day running variance of intraseasonal currents normal to the section. (c) First EOF mode of the seasonal cycle of the DO eddy flux normal to the oblique section depicted in Fig. 17 by the dashed blue line (Southern boundary). (d) PC timeseries associated with the first EOF mode (black line). The blue curves (full and dashed lines) in (d) corresponds to the PC timeseries associated with the first and second EOF modes of the seasonal cycle of the 30-day running variance of the intraseasonal currents normal to the section (computed as in (b)). Percentage of explained variance and RMS value are indicated in parentheses in the panels (b) and (d) (in cm s$^{-1}$ µM and cm s$^{-1}$, for DO eddy flux and currents respectively). White contours in (a) and (c) denote mean DO concentration values, in µM. (e) RMS of the spatial patterns (a) and (c), computed along the horizontal direction. Note the scale leap at 300 m. Red/blue shading in (b), (d) and (e) represents an estimate of the error associated with slight changes in the location of the boundaries, that is when the EOF is performed over a section that is located at a distance from the original section (cf. Figure 17) compromised between +/-120km (see text). The error corresponds to the standard deviation among 12 PC timeseries (for (b) and (d)) and EOF patterns (for (e)).

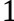1

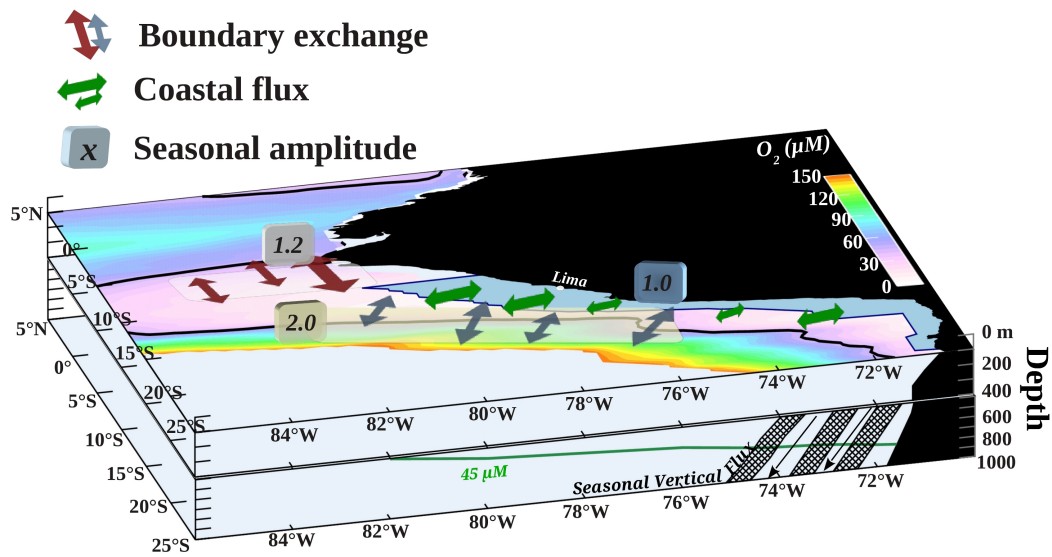

**Figure 20.** Schematic of the main processes driving the seasonal variability in the SEP OMZ: The DO
eddy flux through the northern-southern boundaries and the DO flux that takes place at the coastal
boundary of the OMZ. The coastal band limits are defined by the light blue shading adjacent to the
coast. A scale of the seasonal amplitude of the eddy driven DO flux at each OMZ boundary is indicated
(units in cm s$^{-1}$ µM). The mean DO concentration (color shading) and the position of the 45 µM
isopleth (thick black contour) at 100 m depth are also represented. The vertical/offshore DO flux
induced by the propagation of the annual ETRW across the 45µM isopleth at 25°S is represented in the
bottom panel.

