# Peer review of "Seasonal Variability of the Oxygen Minimum Zone off Peru in a"

_Biogeosciences, 2015_

## Referee Comment (RC1) · Anonymous Referee #1 · 17 Feb 2016

Review of „Seasonal Variability of the Oxygen Minimum Zone off Peru in a high-resolution regional coupled model" by O. Vergara, B. Dewitte, I. Montes, V. Garçon, M. Ramos, A. Paulmier, and O. Pizarro.

The study focuses on the seasonal oxygen variability and the oxygen budget of the South Pacific Oxygen Minimum Zone (OMZ). It reports interesting and new results regarding important tropical ocean variability. The manuscript is largely based on a high-resolution physical/biochemical model simulation that are very well validated against available observations. With this model simulation the oxygen budget on seasonal timescales is evaluated, particularly focusing on the role of physical and of biogeochemical processes for the seasonal oxygen changes. The authors are able to define and calculate the relative importance of physical and biogeochemical processes in different regions and water depths, which represents an important result for our understanding of the annual cycle of oxygen. The manuscript is in general well written and represents a very valuable contribution to the existing literature. However, part of the analysis and particularly figures and its captions must be clearer to be completely understandable. I will give some suggestions and hints for consideration.

General Points:
The term mixing used throughout the paper is not well defined from the beginning. Does it refer to lateral (eddy-induced) or diapycnal mixing or both. Or does it correspond to the $3^{rd}$ term of the rhs of equation 1? Please clarify. Also the lateral mixing term in equation 1 ($2^{nd}$ term rhs) is not well defined. Does it describe the subgrid-scale diffusivity? What is $K_h$ in the model? Is it evaluated as part of the oxygen budget? Is it not important?

Fig. 12 seems to be a main result, but is very difficult to understand. It shows on the one hand the mean oxygen and energy flux resulting form the propagation of annual ETRW (Fig. 12a,b) and on the other hand the annual cycle of the DO eddy flux. It would be good to separate these 2 topics and discuss them separately. In this way you could add also phase information in a new panel to Fig. 12c, which would make the annual cycle of DO eddy flux better understandable.

Specific points:
P2, L16: "… DO eddy flux dominates over the mean seasonal DO flux ..."
It is not so clear what is meant: DO eddy flux includes a mean and a seasonal cycle; mean seasonal DO flux (is also not defined and difficult to understand) can also include a mean and a seasonal cycle. What "dominates" refer to?

P2, L19: what is "mean eddy flux" here?

P2, L22: Implications of the results …

P3, L5: presently

P3, L8: "climatic gases" sounds awkward and not correct: better "greenhouse gases"?

P3, L9: "sequestration ocean role", please reformulate

P5, L6: reduce … the offshore transport (of what?)

P5, L7 also P5, L20: "eddy-induced" instead of "eddies-induced"

P5, L10-12: mesoscale activity can only be a ventilation process if the higher oxygen is found at the coast, which might be present during some time of the year. What are OMZ properties? Typically, I would assume that OMZ properties refer to minimum oxygen; the transport of it would not represent a ventilation of the offshore OMZ. Please clarify.

P8, L4: "OMZ equilibrium is reached": water mass ages are typically much older in OMZs compared to the length of the used simulation. Thus the equilibrium is probably not reached. However, the simulation can still be quite stable and drifting not much away from the initial conditions, which should be good for the used application of the model.

P8, L20/21: something is missing like "obtained " or similar

P9, L1 and L9: avoid the repetition

P10, L5: Here you have to define exactly the used terms: e.g., mean seasonal DO flux (also mentioned in the abstract), DO flux obtained from annual harmonics (Fig. 12), etc.; please be careful and use the defined terms throughout the manuscript. What is a mean flux and what is a seasonally varying flux?

P10, L5: I am not sure to what "latter" refers to.

P10, L8: How is the departure from monthly mean calculated? Just subtracting from the time series monthly means during each month would result in step-like functions. Do you use a running mean or any other kind of interpolation?

P10, L13: How is $K_z$ and $K_h$ defined in the model.

P10, L16: How large is $K_h$? Is the horizontal diffusion further used in the oxygen budget, e.g. Fig 11b? What is mixing in Fig. 11b (on P5, L7 and L20 you use the term mixing for eddy-induced mixing)? What represents the residual in Fig. 11b?

P11, L3-6: Please clarify here, why do you need baroclinic mode decomposition to derive WKB ray paths. According to Ramos et al. (2008), in the long wave approximation the WKB ray path depend on the local vertical wave number. Such vertical wavenumber could be derived from the vertical distribution of the annual harmonic amplitude and phase. Please explain, why do you use instead phase speed values from different baroclinic modes and how this is in agreement with the WKB assumptions?

P11, L15: With the given normalization it is difficult to understand the amplitude of the annual harmonic. It would help to also see the harmonic amplitude without normalization.

P12, L4: at this stage it is not clear what is meant with advection and mixing. What is about eddy fluxes, and horizontal diffusion?

P13, L19: What does (deoxygenated) mean in brackets next to oxygen?

P14, L6: What is mixing here?

P14, L27-30: What is the constant input of low oxygen waters? This should be mean advection. Please clarify or reformulate.

P16, L11: Mean seasonal flux: My understanding is that Fig 12a shows a contribution of annual oscillations to the mean oxygen budget as it is averaged over time. This is very different from the remaining part of the section "Mean seasonal flux". Fig12c shows the annual oscillation of the DO eddy flux. Please clarify why these two terms are discussed together?

P16, L13: Is the seasonal DO flux maximum below 400 m? It is difficult to see as in Fig 12a as DO is normalized by the variance of its climatology. How does it compare to the contribution of the DO eddy flux to the mean budget?

P16, L17: "annual energy flux vector". Please be consistent throughout the manuscript with the use of annual" and "seasonal". Here (Fig. 12b) annual is used, while for Fig. 12a seasonal is used. Annual is typical used for annual harmonics, while seasonal might be used for the mean seasonal cycle.

P17, L3: "thus" seems awkward at the beginning of a new section.

P17, L4,5: Offshore transport of DO also depends on the DO gradient, which can be directed onshore or offshore (see e.g. Fig. 2c) and might vary on seasonal time scales.

P19, L17: latitudinal variability: variability of what? I am not sure what is meant: variability of meridional velocity?

P19, L18,19: I don't see the connection: Why indicate the existence of eddies that they can remove DO from the OMZ, which represents a up gradient flux? Can the DO eddy flux become negative or only anomalous weak?

P19, L24: "DO exhibits … amplitude" seems not to be correct. I would suggest: … regions with enhanced amplitude or specific propagation characteristics, suggesting …

P20, L8: upper 300 m.

P21, L6: suggestion: …Anticyclone, and the peak in the seasonal DO eddy flux coincides with the reported …

P21, L9: Similar processes were discussed in Qiu et al. (2013, JPO).

P21, L20: please specify the range of frequency that you have in mind

P31, Figure 1 and following figures: if used term „depth" in contour labels or at the axes it should be always positive

P32, L32: mean zonal speed

P34, L5: what is total? Please define.

P36, phase figure could be in color to better see the different regions/phase propagation

P36, L4-5: "ray paths for a baroclinic mode" seems not correct, as a single baroclinic mode cannot produce a ray path.

P36, L6: oxygen not capital

P38, L38: nice figure

P39, caption: Solid white lines (c) denote ….

P40: Is horizontal diffusion from equation 1 included in one of these terms or is it neglected?

P41, L5,6: It is not clear, how to reconstruct the original fields from the EOF pattern, the principle components and RMS. Which terms are dimensional, which are non-dimensional, how is the RMS calculated, has the RMS units?

P41, L6: definition (region, width of the band along the coast, etc.) of alongshore winds, coastal sea level, Chl- a and MLD are required.

P42, Fig. 12: This figure is very difficult to understand and addresses different topics (so far I can understand). From the given formulas, I would assume that Figs. 12a and b represent mean fluxes. Fig. 12a represents a contribution to the mean oxygen budget (however, the importance of such a term relative to other terms of the budget is unknown). Fig. 12c represents the annual cycle of the DO eddy flux corresponding to an annually oscillating on- and offshore DO flux (is this correct?). Why are these terms together in one figure? Other points regarding this Figure: a) units are not given, taken into account the normalization, it should be m/s. Is this correct? b) why is the unit m^2/s, how is it calculated? Arrows indicate vector direction and strength. For Fig. 12c a phase information would be helpful.

P42, L11: What is DO: $O_2$' or $O_2^{1yr}$. It is not clear, how the normalization is performed. I would suggest to be consistent throughout the manuscript in using either DO, $O_2$, oxygen concentration etc.

P43, L6: delete "average profile"?

P44, L5: direction and strength

P44, L5: include "mean oxygen values"

P45, L5: denote mean oxygen

P46, figure caption: very difficult to understand. Please clarify and better specify what is shown. Be consistent "dominant EOF" and "first EOF". L6: "running variance of currents (dominant EOF mode)": I am not sure what is meant. L7: "… section (blue line in Fig. 14)". L13: "Dispersion" is very unusual term and I would suggest to find a better description of it. Are the spatial pattern normalized before calculating RMS and standard deviation.

P47, Fig. 17: Why is the oxygen flux from the coastal boundary into the OMZ not back and forth as the eddy fluxes at the northern-southern boundaries? Fig. 12c would suggest this. L7: The position … at which depth? Please also explain the green line, where it is calculated. Again, this schematic seems to mix mean and seasonally varying fluxes. Is this the aim?

---

## Referee Comment (RC2) · Anonymous Referee #2 · 8 Mar 2016

Biogeosciences review:

Summary: A high-resolution regional physical/biogeochemical model is used to examine the seasonal and interannual controls over the variability of the OMZ in the south eastern pacific. The competing influence of mesoscale activity in particular is explored. Specifically the role of the mesoscale activity on the OMZ in reducing biological production vs. providing increased ventilation – both would increase oxygen in the region. While the work is interesting and the question a good one – the work is muddled by lack of organization, clarity of approach, and grammar issues. The authors would greatly improve this manuscript by organizing it in a fashion something like – 1) problem description in the introduction – clearly stating issue of competing roles of

mesoscale activity, 2) approach/methods used including specific details about analysis, 3) validation of mesoscale activity in the model (a section currently missing), 4) results beginning with the oxygen budget, bio vs physical, and moving into breaking it into subregions (this approach the authors took of subregional discussions was a good idea), 5) discussion of the possible mechanisms to explain these results and primary means of ventilation to the OMZ 6) conclusions. The discussion and results sections are muddied together currently. Without a clear distinction, the reader becomes lost. This work would benefit from some reorganization, clarifying elements in the figures, and a table with the budget analysis more clearly laid out. The biggest concern is the lack of validation of the model fields for the processes identified in this work as being important – the seasonal cycle, or the mesoscale activity, etc. Prior work is cited for validation, but it focused on an annual climatology.

Major comments: The additional section regarding the validation of the mesoscale activity within the model seems critical to identifying this mechanism. Specifically, the validation referenced in Gutknecht et al. 2013 seems to focus on validating the mean conditions, and Figure 6 of that work indicates some clear subsurface biases that need to be taken into account for these mechanisms. How do comparisons to the AVISO data set look? What about temperature? Can you rule out advection and confirm that the bias is in the biogeochemical model? The bias needs to be put in context – if the authors can describe the bias and then put their results in context that would provide much more information to the reader. The validation provided here is not only qualitative, but it also focuses on the average conditions. Some indication at to how well the model does at representing the variability and fluxes is necessary. Fluxes can include the SMS terms – like production. It is common now for taylor or target diagrams to be used to visualize the metrics of skill, this work would benefit from one that may also include some physical terms and goes beyond the climatology.

Also, the model climatology is an average over 10 years, while the data climatology is over 50 years. It seems a better apples to apples comparison could be made.

The budget analysis is a powerful tool and has a lot of information within it. The authors appropriately use this tool to try and tease out the relative contribution of biological processes to the physical processes. It seems necessary to show the budget balances in the regions that this tool is used to discern the relative processes. A table of the budget terms would be one suggested way to achieve this goal. What does the climatological base budget look like? How big are the anomalies and residuals focused on in this work?

The methods section ends with a paragraph that seems more like discussion - describing vertical model decomposition. This doesn't seem to be very well woven into the rest of the paper and comes off as a bit of a distraction or outlier.

Given the paper's premise that the high resolution study provides further information than a course simulation would, some comparison seems warranted showcasing that result.

Some of the framing in the abstract about greenhouse gases is isolated there. While it is interesting, the authors never return to that idea in the discussion. Either return to it or remove those ideas.

Minor technical comments:

Lines 10-12 on Page 9 of the introduction points blame at the resolution of the CARS data set – the authors could add this resolution to some of their figures in order to better make this point, but without proper validation of the model, it is hard to blame the observations.

Line 4 on Page 3 – in introduction – understand should be understanding.

Lines 10-11 on Page 3 in intro – The sentence beginning with Furthermore refers to a process called "habitat compression". The phrase may be more clear than the current explanation.

Line 12 page 16 – Relatively should be Relative.

[Figure]

Figure comments: The main comment in all the figures from the model fields is about the white space – what does it indicate?

Figure 1 seems to indicate the model does not achieve the same onshore/offshore gradients in oxygen around 10 deg S. This seems critical to some of the points in the paper about transport mechanisms and goes unaddressed in the text.

Figures 1-3 would benefit from additional dots on the figures to identify where the samples the made up the climatology from the observations were made. The reader could discern from that addition the errors in interpolation to errors in the model.

Figure 4 would benefit from isobaths contours so the shelf region was highlighted.

Figure 5 seems to indicate the model is biased high – that the model underestimates the hypoxic volume. Is that because of advection, temperature, or the bio model? Or could it be the different time periods compared between the model and obs?

Figure 6 – this figure and discussion are out of context and need to be reorganized at a more appropriate time in the results. These numbers need to be put in context with the other important fluxes as you do in the budget. So this needs to follow the budget.

Figure 8 a and b panels do not appear to be on the same scale. Refer to Equation 2 from the text in the caption.

Figure 10 is confusing but important. Clarity to the reader would be achieved through a longer caption explaining the different colorbars in a-c and d-g, as well as the percentages – which don't add up. Why is the undercurrent identified? Is it referred to in the text? Are these all model results? What transect is this in the domain? Does the choice of the transect change the results? The text refers to this figure describes seasonality, but this figure just says climatology – is it a seasonal climatology? In that case, what time of year is shown?

Figure 11 requires more explanation as to how it was made. No explanation of residuals etc is provided in the text. How should the reader interpret the residual? Is it meant to

just be physical?

---

## Author Comment (AC1) · 3 May 2016

Please find the responses to the referees' comments in the attached file below.

Please also note the supplement to this comment:
http://www.biogeosciences-discuss.net/bg-2015-631/bg-2015-631-AC1-supplement.pdf

---

## Author Comment (AC2) · 3 May 2016

**Response to Anonymous Referee #1**

**General points**

We would like to thank the reviewer for his/her thorough review and constructive comments. We have addressed all of them and provide below a detailed reply.

**Comment 1 (C1).** The term mixing used throughout the paper is not well defined from the beginning. Does it refer to lateral (eddy-induced) or diapycnal mixing or both. Or does it correspond to the 3rd term of the rhs of equation 1? Please clarify. Also the lateral mixing term in equation 1 (2nd term rhs) is not well defined. Does it describe the subgrid-scale diffusivity? What is Kh in the model? Is it evaluated as part of the oxygen budget? Is it not important?

**Response (R.):** The reviewer is right. We have clarified what we refer to as mixing in the methods section (Section 2.3). Mixing corresponds in our study to the contribution of horizontal diffusion ( $K_h \cdot \nabla^2 O_2$ ), vertical mixing ( $\frac{\partial}{\partial z}\left(K_z \frac{\partial O_2}{\partial z}\right)$ ) and the mixing that is associated with the non-linear advection. The latter corresponds to $\left(u'\partial u'/\partial x\right)+\left(v'\partial v'/\partial y\right)+\left(w'\partial w'/\partial z\right)$ , assuming the Reynolds decomposition for the velocity field, i.e. $\vec{u}+\vec{u}'$ , where $\vec{u}'$ accounts for the intraseasonal variability (periods lower than ~3 months).

In the model, the lateral mixing term in equation 1 refers to the subgrid-scale diffusivity. $K_h$ corresponds to the eddy diffusivity coefficient, and it is a constant value in that version of ROMS (version 2.1), equal to 100 $m^2 s^{-1}$. Note that the model also has a numerical diffusion associated with inherent spurious diapycnal mixing of the numerical scheme, so that $K_h$ is empirically adjusted, and the term $K_h \cdot \nabla^2 O_2$ does not account for all the actual diffusion in the model.

As suggested by the reviewer, an explanation of the term mixing used in the analyses was added to the paper (Section 2.3).

**C2.** Fig. 12 seems to be a main result, but is very difficult to understand. It shows on the one hand the mean oxygen and energy flux resulting form the propagation of annual ETRW (Fig. 12a,b) and on the other hand the annual cycle of the DO eddy flux. It would be good to separate these 2 topics and discuss them separately. In this way you could add also phase information in a new panel to Fig. 12c, which would make the annual cycle of DO eddy flux better understandable.

**R.** (P55-56): Following the reviewers' recommendation, we now present two figures: one figure with the amplitude of the annual cycle of the climatological DO eddy flux, with the corresponding phase diagram (new Figure 15, see Figure A1) and the figure showing the results on the annual ETRW flux (new Figure 14).

[Figure]

**Figure A1**. Zonal section of the annual harmonic of the module of the seasonal DO eddy flux vector $\left(\overline{\langle u' \cdot O_2' \rangle}, \overline{\langle w' \cdot O_2' \rangle}\right)$ at 12°S. (a) Amplitude of the harmonic and (b) phase of the annual maximum (in months). Dashed white contours indicate the 45 and 20 µM mean DO isopleths. DO was normalized by its RMS prior to carrying out analysis.

**Specific points**

**C1.** P2, L16: "… DO eddy flux dominates over the mean seasonal DO flux …" It is not so clear what is meant: DO eddy flux includes a mean and a seasonal cycle; mean seasonal DO flux (is also not defined and difficult to understand) can also include a mean and a seasonal cycle. What "dominates" refer to?

**R.** (P2,L16): The reviewer is right. We meant that the mean DO eddy flux (i.e. $\langle u' \cdot O_2' \rangle$ ) is larger over the mean seasonal DO flux $\langle \tilde{u} \cdot \tilde{O}_2 \rangle$ , where the fields with tilde stand for the seasonal component (see also C13 in Specific points). We have also clarified the term "dominate" and changed the sentence to: "…an upper zone above 400m where mean DO eddy flux is larger on average than the mean seasonal DO flux".

**C2.** P2, L19: what is "mean eddy flux" here?

**R.** (P2, L20): This was changed to "mean DO eddy flux", and the sentence was modified accordingly.

**C3.** P2, L22: Implications of the results …

**R.** (P2, L24): The sentence was corrected.

**C4.** P3, L5: presently

**R.** (P3, L6): The sentence was corrected.

**C5.** P3, L8: "climatic gases" sounds awkward and not correct: better "greenhouse gases"?

**R.** (P3, L8): This was corrected using "climatically-active gases".

**C6.** P3, L9: "sequestration ocean role", please reformulate

**R.** (P3, L10): The sentence was changed to "...ocean carbon sequestration..."

**C7.** P5, L6: reduce ... the offshore transport (of what?)

**R.** (P5, L9): We meant offshore transport of carbon. The sentence was changed to "... offshore carbon export..."

**C8.** P5, L7 also P5, L20: "eddy-induced" instead of "eddies-induced"

**R.** (P5, L10): The sentence was corrected.

**C9.** P5, L10-12: mesoscale activity can only be a ventilation process if the higher oxygen is found at the coast, which might be present during some time of the year. What are OMZ properties? Typically, I would assume that OMZ properties refer to minimum oxygen; the transport of it would not represent a ventilation of the offshore OMZ. Please clarify.

**R.** (P5, L14): We refer here to the exchange of tracers between the shelf and the open ocean induced by the mesoscale activity, and the OMZ properties refer to oxygen, organic carbon and/or nitrate. It was implicitly assumed that there exists a cross-shore gradient in water mass properties, so a transport can actually take place. The text has been clarified as follows: "In this sense, the mesoscale activity represents a ventilation pathway for the OMZ, through the offshore transport of oxygen-enriched waters".

**C10.** P8, L4: "OMZ equilibrium is reached": water mass ages are typically much older in OMZs compared to the length of the used simulation. Thus the equilibrium is probably not reached. However, the simulation can still be quite stable and drifting not much away from the initial conditions, which should be good for the used application of the model.

**R.** (P9, L2): The sentence was changed to: "Although, after the spin-up, the simulation has reached stable conditions and the OMZ volume does not drift, we focus in the present study only on the period 2000-2008".

**C11.** P8, L20/21: something is missing like "obtained " or similar.

**R.** (P9, L24)**:** Added "computed". The sentence was corrected to: "...with $O_2$sat computed following the methodology of Garcia and Gordon (1992)".

**C12.** P9, L1 and L9: avoid the repetition

**R.** (P10, L5): The sentence corresponding to L1 was changed to: "...the simulation presents more details than the climatological product" in order to avoid the repetition.

**C13.** P10, L5: Here you have to define exactly the used terms: e.g., mean seasonal DO flux (also mentioned in the abstract), DO flux obtained from annual harmonics (Fig. 12), etc.; please be careful and use the defined terms throughout the manuscript. What is a mean flux and what is a seasonally varying flux?

**R.** (P11, L14): In order to clarify the definition of the different fluxes, we have added the mathematical formula each time we define a flux, and the adjective "mean" was also added when required in order to clarify. The text in the revised manuscript is now: "The DO flux associated with different timescales of variability is therefore estimated. This consists in computing the temporal average of the cross-products between DO and velocity anomalies. Anomalies can refer either to seasonal anomalies and in that case, this provides the mean seasonal DO flux ( $\langle \tilde{u} \cdot \tilde{O}_2 \rangle$ where ~ refers to the seasonal anomalies), or to the intraseasonal anomalies (calculated here as the departure from the monthly mean) and in that case, this provides an estimate of the mean DO eddy flux ( $\langle u' \cdot O_2' \rangle$, where the apostrophe refers to the intraseasonal anomalies). In this paper we are also interested in the seasonality of the DO eddy flux. This is estimated from the monthly-mean seasonal cycle of the mean DO eddy flux calculated over a 3-month running window, and is now referred to as $\overline{\langle u' \cdot O_2' \rangle}$ ".

**C14.** P10, L5: I am not sure to what "latter" refers to.

**R.** (P11, L21): It refers to the estimation of DO flux associated with different scales of variability. The sentence was reformulated in order to avoid confusion: "The DO flux associated with different timescales of variability is therefore estimated. This consists in computing the temporal average of the cross-products between DO and velocity anomalies".

**C15.** P10, L8: How is the departure from monthly mean calculated? Just subtracting from the time series monthly means during each month would result in step-like functions. Do you use a running mean or any other kind of interpolation?

**R.** (P11, L24): In order to derive intraseasonal anomalies, we first compute the monthly mean data. We then interpolate the monthly mean data on the original time grid (3-day mean) using a cubic spline, and use this as a baseline: the anomalies are obtained from the difference between the original total field (direct model output) and the interpolated monthly mean. This methodology has been used in previous studies (Lin et al., 2000; Dewitte et al., 2011) to study intraseasonal variability in surface winds and SST. This method is similar to a high-pass filter with a transfer function, estimated by computing intraseasonnal anomalies of a daily Gaussian white noise, characterized by a −1, −3,and −10 dB attenuation (79%, 50%, 10% of the input power persists) at 59, 68, and 96 days$^{-1}$ frequency, respectively. This would be also equivalent to a Lanczos filter with a cut-off period of 90 days.

References:

Dewitte, B., Illig, S.,Renault, L., Goubanova, K., Takahashi, K., Gushchina, D., Mosquera, K., and Purca, S.: Modes of covariability between sea surface temperature and wind stress intraseasonal anomalies along the coast of Peru from satellite observations (2000–2008), J.Geophys. Res., 116, C04028, doi:10.1029/2010JC006495, 2011.

Lin, J. W. B., Neelin, J. D., and Zeng, N.,: Maintenance of tropical intraseasonal variability: Impact of evaporation–wind feedback and midlatitude storms, J. Atmos. Sci., 57, 2793–2823, doi:10.1175/1520-0469, 2000.

**C16.** P10, L13: How is Kz and Kh defined in the model.

**R.** (P12, L6): In this version of ROMS (version 2.1), $K_h$ is fixed to 100 m$^2$ s$^{-1}$.

$K_z$ is calculated by the vertical mixing scheme which is based on the KPP scheme (Large et al., 1994).

References:

Large, W. G., McWilliams, J. C., and Doney, S. C.: Oceanic vertical mixing: A review and a model with a nonlocal boundary layer parameterization, Rev. Geophys., 32, 363–403, doi: 199410.1029/94RG01872, 1994.

**C17.** P10, L16: How large is Kh? Is the horizontal diffusion further used in the oxygen budget, e.g. Fig 11b? What is mixing in Fig. 11b (on P5, L7 and L20 you use the term mixing for eddy- induced mixing)? What represents the residual in Fig. 11B?

**R.** (P12, L7): In this version of ROMS (version 2.1), $K_h$ is fixed to 100 m$^2$ s$^{-1}$.

In figure 11b, the term mixing refers just to the contribution of both vertical and horizontal mixing terms from Eq. 1. We have changed the notation in Figure 10g and 11b (Fig. 12g and 13b in the revised version of the manuscript) and now use the notation: Hmix+Vmix, where Hmix and Vmix refer to horizontal diffusion and vertical diffusivity respectively.

The residual in Figure 13b corresponds to the difference between the rate of DO change and the sum of all the terms on the rhs of Eq. 1 for the normalized PC timeseries. Our purpose is to verify that the DO budget is nearly closed when considering the first EOF modes of each term. Although the residual is not zero, it remains weak (two orders of magnitude smaller than the PCs of the DO budget terms), allowing for the interpretation of the different terms using their first EOF modes.

Modifications were introduced to the caption of Figure 11 (Figure 13 in the revised version of the manuscript) in order to clarify: "The residual corresponds to the difference between the rate of DO change and the sum of all the terms of the rhs of Eq. 1 in terms of the normalized PC timeseries. The weak residual indicates that the seasonal DO budget can be interpreted from the EOF decomposition."

Note that we have also clarified the caption of Figure 10 (Figure 12 in the revised version of the manuscript): "The EOF mode patterns were multiplied by the RMS of the PC timeseries. Multiplying the EOF pattern by the PC timeseries plotted in Figure 13 yields the contribution of the first EOF mode to the original field, in dimensionalized units (i.e. $\mu M\ s^{-1}$ for the tendency terms)".

**C18.** P11, L3-6: Please clarify here, why do you need baroclinic mode decomposition to derive WKB ray paths. According to Ramos et al. (2008), in the long wave approximation the WKB ray path depend on the local vertical wave number. Such vertical wavenumber could be derived from the vertical distribution of the annual harmonic amplitude and phase. Please explain, why do you use instead phase speed values from different baroclinic modes and how this is in agreement with the WKB assumptions?

**R.** (P13, L4-9): The slope of ray paths in the (x, z) plane is $\frac{dx}{dz} = 2\frac{\omega f^2}{\beta c N}$ (see Appendix in Ramos et al. (2008)) where c is the "observed" phase speed of the Extra-tropical Rossby wave which consists in a superposition of different baroclinic modes, so the speed can range from values $c_1$ to $c_n$, where $c_n$ is the theoretical phase speed of a $n^{th}$ baroclinic mode, obtained from the vertical mode decomposition of the local density profile. We have clarified the text, mentioning that the vertical mode decomposition is only aimed at providing a range of values for c, consistent with the model stratification.

References:

Ramos, M., Dewitte, B., Pizarro, O., and Garric, G.: Vertical propagation of extratropical Rossby waves during the 1997 – 1998 El Niño off the west coast of South America in a medium-resolution OGCM simulation, J. Geophys. Res., 113, C08041, doi:10.1029/2007JC004681, 2008.

**C19.** P11, L15: With the given normalization it is difficult to understand the amplitude of the annual harmonic. It would help to also see the harmonic amplitude without normalization.

**R.** (P13, L26): The objective of the DO normalization is to highlight the regions where the DO concentration is weak. We expect to have a weak annual cycle where the DO is very low (see Figure

A2). By doing the normalization, the phase pattern is unaltered but the spatial variation of the annual cycle amplitude is emphasized.

[Figure]

**Figure A2**. (a) Amplitude and (b) phase (in months) of the DO annual harmonic over a zonal section at 12°S (without normalization). Dashed lines in (a) correspond to the 20 et 45 µM mean DO isopleths.

**C20.** P12, L4: at this stage it is not clear what is meant with advection and mixing. What is about eddy fluxes, and horizontal diffusion?

**R.** (P14, L15): At this stage of the paper, we first diagnose the DO budget using the Eq. (1). Therefore, advection refers to the first rhs term in that equation, and mixing refers to the sum of the lateral and vertical mixing terms from the same equation (2nd and 3rd rhs terms in Eq. (1)). We are now referring to the Equation 1 for this sentence, that was added to the paper: "Advection corresponds to the first term on rhs of Eq. (1), whereas mixing refers to the sum of the lateral and vertical mixing terms (2nd and 3rd rhs terms in Eq. (1))".

We have added arrows on to top of the parameters in Eq. 1 in order to clarify:

$$\frac{\partial O_2}{\partial t} = -\vec{u} \cdot \left( \vec{\nabla} O_2 \right) + K_h \nabla^2 O_2 + \frac{\partial}{\partial z} \left( K_z \frac{\partial O_2}{\partial z} \right) + SMS\left( O_2 \right)$$

**C21.** P13, L19: What does (deoxygenated) mean in brackets next to oxygen?

**R.** (P15, L29): A word was missing in this sentence. It was replaced by: "...the DO exchange between the coastal domain and the OMZ takes place through the offshore transport of DO poor waters by eddies (Czeschel et al., 2011)...".

**C22.** P14, L6: What is mixing here?

**R.** (P16, L24): Mixing refers here to "total" mixing that is: horizontal diffusion + vertical mixing + mixing associated with non-linear advection (see C1 in General Comments). Later on in the paper, unless stated otherwise, the term mixing alone (i.e. without specifying horizontal and/or vertical) will always refer to that. This has been clarified in the revised version of the manuscript (Section 2.3).

**C23.** P14, L27-30: What is the constant input of low oxygen waters? This should be mean advection. Please clarify or reformulate.

**R** (P17, L18-19): The reviewer is right. *Constant input* refers to the mean advection by the PUC. The sentence was changed to: "...balanced out by the mean advection of low DO waters carried by the PUC..."

**C24.** P16, L11: Mean seasonal flux: My understanding is that Fig 12a shows a contribution of annual oscillations to the mean oxygen budget as it is averaged over time. This is very different from the remaining part of the section "Mean seasonal flux". Fig12c shows the annual oscillation of the DO eddy flux. Please clarify why these two terms are discussed together?

**R** (P19, L1): Following the reviewer's recommendation we have clarified the text. By averaging the cross-product of the annual harmonics of velocity and DO, we obtain the contribution of the annual harmonic to the seasonal DO flux (i.e. $\langle u^{1yr} \cdot O_2^{1yr} \rangle$ ), which we compare to the annual harmonic of the DO eddy flux in order to highlight the different amplitude patterns. Our objective is here to identify the regions of influence of the different timescales of variability (annual versus intraseasonal) in terms of DO flux. For clarity we now present two figures: the Figure 14 for the mean annual flux (for DO and momentum) and the Figure 15 for the amplitude and phase of the annual DO eddy flux (cf. C2 in General Points). The text of the paper was modified accordingly.

**C25.** P16, L13: Is the seasonal DO flux maximum below 400 m? It is difficult to see as in Fig 12a as DO is normalized by the variance of its climatology. How does it compare to the contribution of the DO eddy flux to the mean budget?

**R** (P19, L5): Below 400m, the seasonal DO flux reaches values around 10% to 40% of those found around the depth of the oxycline, as illustrated in Figure A3a (which is similar to Fig. 14a of the revised version of the manuscript, but without normalization). At this location, the contribution of the annual DO flux is around 10 times larger than the contribution of the mean zonal eddy DO flux (Fig. A3b).

[Figure]

**Figure A3.** (a) Norm of the annual DO flux vector (i.e. $\sqrt{\left(\langle u^{1\text{yr}}\cdot O_2^{1\text{yr}}\rangle\right)^2+\left(\langle w^{1\text{yr}}\cdot O_2^{1\text{yr}}\rangle\right)^2}$ ) for a cross shore section at 12°S. The superscript "1yr" refers to the contribution of the annual harmonic. (b) Mean zonal DO eddy flux along the same section. Dashed black contours indicate the 45 and 20 µM oxygen isopleths. Units are in cm s⁻¹ µM.

**C26.** P16, L17: "annual energy flux vector". Please be consistent throughout the manuscript with the use of annual" and "seasonal". Here (Fig. 12b) annual is used, while for Fig. 12a seasonal is used. Annual is typical used for annual harmonics, while seasonal might be used for the mean seasonal cycle.

**R** (P19, L9): "Seasonal" was changed for "annual" when addressing the Figure 14a (Figure 12a in the first version of the manuscript). We checked for consistency throughout the paper and have done the necessary changes to the text.

**C27.** P17, L3: "thus" seems awkward at the beginning of a new section.

**R** (P19, L26): The sentence was changed to: "As previously described, the annual amplitude of the climatological DO eddy flux is the largest in the upper 400 m near the coast"

**C28.** P17, L4,5: Offshore transport of DO also depends on the DO gradient, which can be directed onshore or offshore (see e.g. Fig. 2c) and might vary on seasonal time scales.

**R** (P19, L18-30): The reviewer is right. The offshore transport will depend on the DO gradient direction. We have changed the sentence: "Since EKE is large along the coast of Peru, exchange of DO induced by eddies could be expected at all latitudes, with a direction that depends on the sign of the DO gradient at the coast".

**C29.** P19, L17: latitudinal variability: variability of what? I am not sure what is meant: variability of meridional velocity?

**R** (P22, L17): This was a mistake. The correct sentence is: "At both boundaries, the zonal wavelength of the seasonal eddy flux variability along the boundary is estimated to be of the order of ~$10^2$ km, a scale that falls within the range of observed eddies diameter (Chaigneau and Pizarro, 2005), which indicates that locally there can be an injection or removal of DO across the boundary on average over a season".

**C30.** P19, L18,19: I don't see the connection: Why indicate the existence of eddies that they can remove DO from the OMZ, which represents a up gradient flux? Can the DO eddy flux become negative or only anomalous weak?

**R** (P22, 20): The Figure 19 (Figure 16 in the first version of the manuscript) indicates that locally the DO eddy flux can be either positive or negative depending on the season, so that there can be a local injection or removal of DO across the boundary. The sentence has been clarified (see C29).

**C31.** P19, L24: "DO exhibits ... amplitude" seems not to be correct. I would suggest: ... regions with enhanced amplitude or specific propagation characteristics, suggesting …

**R** (P28, L27): The sentence was changed following the suggestion of the reviewer: "The annual harmonic of DO reveals three main regions with enhanced amplitude or specific propagation characteristics, suggesting distinct dynamical regimes".

**C32.** P20, L8: upper 300 m.

**R** (P29, L11): The sentence was changed to: "This appears to operate effectively in the upper 300 m".

**C33.** P21, L6: suggestion: ...Anticyclone, and the peak in the seasonal DO eddy flux coincides with the reported …

**R** (P27, L8-9): The sentence was changed following the suggestion of the reviewer: "and the peak in the seasonal DO eddy flux coincides with the reported intensity peak of the seasonal cycle of the Anticyclone".

**C34.** P21, L9: Similar processes were discussed in Qiu et al. (2013, JPO).

**R** (P27, L11-15): We thank the reviewer for mentioning this interesting paper. We have added this reference to the text of the revised manuscript: "Dewitte et al. (2008) also report that intraseasonal (internal) variability in currents can originate from the interactions between the annual extra tropical

Rossby wave and the mean circulation in a medium resolution oceanic regional model over this region, a process also observed and documented from a high-resolution model over the North Pacific (Qiu et al., 2013)."

**C35.** P21, L20: please specify the range of frequency that you have in mind

**R** (P27, L24-25): We refer to the annual to interannual range. The sentence was changed to: "at frequencies ranging from annual (Dewitte et al., 2008) to interannual (Ramos et al., 2008)".

**C36.** P31, Figure 1 and following figures: if used term „depth" in contour labels or at the axes it should be always positive

**R** (P42): We thank the reviewer for pointing out that mistake. All the figures were modified accordingly.

**C37.** P32, L32: mean zonal speed

**R** (P43, L3): The sentence was corrected.

**C38.** P34, L5: what is total? Please define.

**R** (45, L4): We refer to the total Particulate Organic Carbon (POC) flux at the depth of 100m. We obtain this quantity by integrating the POC flux over the horizontal area in the Figure 5 (Figure 4 in the first version of the manuscript). This definition was added to the figure caption in order to clarify.

**C39.** P36, phase figure could be in color to better see the different regions/phase propagation

**R** (P48): The phase diagram was shaded in gray scale in order to highlight the phase variations.

**C40.** P36, L4-5: "ray paths for a baroclinic mode" seems not correct, as a single baroclinic mode cannot produce a ray path.

**R** (P48, L24-27): We agree with the reviewer. We have modified the manuscript and have clarified that we use the value of phase speed associated with different baroclinic modes to draw the WKB ray paths, in order to estimate the range of trajectories that WKB ray paths can take at a given frequency.

The caption of Figure 8 (Figure 6 in the first version of the manuscript) was changed to: "The slanted vertical lines indicate the theoretical WKB ray paths at a frequency of $\omega=2\pi\cdot1\text{year}^{-1}$, for different value of phase speed. The theoretical trajectories were computed using the phase speed of the first (full), second (dashed) and third (dotted) baroclinic modes of a long Rossby wave."

**C41.** P36, L6: oxygen not capital

**R** (P48, L29): Corrected.

**C42.** P39, caption: Solid white lines (c) denote ....

**R** (P51, L3): This was corrected.

**C43.** P40: Is horizontal diffusion from equation 1 included in one of these terms or is it neglected?

**R** (P52): Caption and title of Figure 12g (Figure 10g in the first version of the manuscript) was clarified. In particular "Mix" was replaced by Hmix+Vmix, in order to clarify that we are referring to horizontal and vertical diffusivity, respectively (i.e. the second and third terms on the rhs of Eq. 1).

**C44.** P41, L5,6: It is not clear, how to reconstruct the original fields from the EOF pattern, the principle components and RMS. Which terms are dimensional, which are non-dimensional, how is the RMS calculated, has the RMS units?

**R** (P53, L11-12): We have clarified this point (see also C17 in Specific Points): Figure 12 (Figure 10 in the first manuscript version) corresponds to the dimensionalized EOF mode patterns. These mode patterns were dimensionalized with (i.e. multiplied by) the RMS of the PC timeseries that are presented in normalized form in Figure 13 (Figure 11 in the first version of the manuscript). So multiplying the patterns by the associated timeseries of Figure 13 provides the reconstructed field (in dimensionalized units). The RMS values are therefore in dimensionalized unit.

The caption of Figure 13 (Figure 11 in the first manuscript version) was clarified as follows: "(a, b) Non dimensional principal components (PC) associated with the EOF patterns in Figure 12. Multiplying the principal component by the associated EOF pattern (from Fig. 12) yields a first EOF-mode reconstruction of the original field. RMS values of the principal components are indicated in parenthesis (corresponding units as in Fig. 12)."

**C45.** P41, L6: definition (region, width of the band along the coast, etc.) of alongshore winds, coastal sea level, Chl- a and MLD are required.

**R** (P54, L2-8): The recommended modifications were introduced to the figure caption: "...coastal alongshore wind (AS wind) and coastal alongshore wind Running Variance (variance over a 30 day running window) at 12°S, sea level at the coast at 12°S, surface chlorophyll-a from CR BIO (Chla) and from SeaWiFS (Chla SW) averaged over a coastal band of 2° width at 12°S, and Mixed Layer Depth at the coast (MLD) at 12°S".

**C46.** P42, Fig. 12: This figure is very difficult to understand and addresses different topics (so far I can understand). From the given formulas, I would assume that Figs. 12a and b represent mean fluxes. Fig. 12a represents a contribution to the mean oxygen budget (however, the importance of such a term relative to other terms of the budget is unknown). Fig. 12C represents the annual cycle of the DO eddy flux corresponding to an annually oscillating on- and offshore DO flux (is this correct?). Why are these terms together in one figure? Other points regarding this Figure: a) units are not given, taken into account the normalization, it should be m/s. Is this correct? b) why is the unit m^2/s, how is it calculated? Arrows indicate vector direction and strength. For Fig. 12c a phase information would be helpful.

**R** (P55, Fig. 14): First in order to clarify, we have divided this figure in two separate figures: The new Figure 14 presents the norm of the annual DO and momentum flux vector, and the new Figure 15 presents the amplitude and phase of annual harmonic of the module of the seasonal DO eddy flux (see also C2 in General Points and C24 in Specific points).

The three terms in Figure 12 of the first manuscript version were initially discussed together in order to contrast the influence of the different timescales of variability on the DO flux (see C24 in Specific points).

The reviewer is correct: with the given normalization the units for the DO flux (Fig. 14a) are indeed m $s^{-1}$. The units were added to the figure.

The units in Figure 14b (Figure 12b in the first manuscript version) correspond to the product of the velocity and pressure, with pressure expressed in m, rather than the usual N $m^{-2}$, which yields $m^2s^{-1}$.

As suggested by the reviewer, a new figure that represents the annual amplitude and phase of the climatological DO eddy flux was included in the paper (cf. C2 in General Points).

**C47.** P42, L11: What is DO: O2' or O21yr. It is not clear, how the normalization is performed. I would suggest to be consistent throughout the manuscript in using either DO, O2, oxygen concentration etc.

**R** (P55): $O_2'$ represents the intraseasonal DO anomaly, whereas $O_2^{1yr}$ is the DO annual harmonic. $O_2$ was replaced by DO throughout the paper in order to be consistent. However, the symbol for oxygen is used in the figure titles. The figure caption was changed in order to clarify how the DO normalization was made. The text for the caption now includes the sentence (Fig. 14): "The DO signal was normalized by its Root Mean Square value before computing the annual harmonic, in order to emphasize the flux patterns where DO concentration is very low".

**C48.** P43, L6: delete "average profile"?

**R** (57, L5): Modified as suggested.

**C49.** P44, L5: direction and strength

**R** (P58, L5): Modified as suggested.

**C50.** P44, L5: include "mean oxygen values"

**R** (P58, L6): The sentence was modified to include this change.

**C51.** P45, L5: denote mean oxygen

**R** (P59, L3): Sentence corrected.

**C52.** P46, figure caption: very difficult to understand. Please clarify and better specify what is shown. Be consistent "dominant EOF" and "first EOF". L6: "running variance of currents (dominant EOF mode)": I am not sure what is meant. L7: "... section (blue line in Fig. 14)". L13: "Dispersion" is very unusual term and I would suggest to find a better description of it. Are the spatial pattern normalized before calculating RMS and standard deviation.

**R** (P60-61): Following the reviewer's suggestion, we have improved the notation. "dominant EOF" was replaced by "first EOF". The running variance of the currents is now explained. It refers to the RMS of current intraseasonal anomalies over a 1-month running window, with the intraseasonal anomalies estimated as the departure from the monthly mean (like for the DO). We then compute the seasonal cycle of this field, and perform an EOF analysis in order to extract the spatial pattern, and the associated timeseries (plotted in Fig. 19b as the red line and in 19d as the full/dashed blue lines).

What we meant by dispersion is the error associated with a slightly different location of the northern and southern boundaries (cf. Figure 17 for the mean position of the boundaries). We selected sections parallel to the original section that are at a distance comprised between +/-120km (every ~20 Km), providing an ensemble of 12 EOF results. This ensemble is used to estimate the error defined as the the standard deviation among 12 PC timeseries (for (b) and (d)) and EOF patterns (for (e)).

We have clarified the figure caption as follows (Fig. 19): "(a) First EOF mode of the seasonal cycle of the DO eddy flux normal to the section depicted in Fig. 17 by the dashed red line (Northern boundary).

(b) Principal component (PC) timeseries associated with the first EOF mode (black line). The red line in (b) corresponds to the PC timeseries associated with the first EOF mode of the seasonal cycle of the 30-day running variance of intraseasonal currents normal to the section. (c) First EOF mode of the seasonal cycle of the DO eddy flux normal to the oblique section depicted in Fig. 17 by the dashed blue line (Southern boundary). (d) PC timeseries associated with the first EOF mode (black line). The blue curves (full and dashed lines) in (d) corresponds to the PC timeseries associated with the first and second EOF modes of the seasonal cycle of the variance of the intraseasonal currents normal to the section (computed as in (b)). Percentage of explained variance and RMS value are indicated in parentheses in the panels (b) and (d) (in cm s$^{-1}$ µM and cm s$^{-1}$, for DO eddy flux and currents respectively). White contours in (a) and (c) denote mean DO concentration values, in µM. (e) RMS of the spatial patterns (a) and (c), computed along the horizontal direction. Note the scale leap at 300 m. Red/blue shading in (b), (d) and (e) represents an estimate of the error associated with slight changes in the location of the boundaries, that is when the EOF is performed over a section that is located at a distance from the original section (cf. Figure 17) compromised between +/-120km (see text). The error corresponds to the standard deviation among 12 PC timeseries (for (b) and (d)) and EOF patterns (for (e))."

**C53.** P47, Fig. 17: Why is the oxygen flux from the coastal boundary into the OMZ not back and forth as the eddy fluxes at the northern-southern boundaries? Fig. 12c would suggest this. L7: The position ... at which depth? Please also explain the green line, where it is calculated. Again, this schematic seems to mix mean and seasonally varying fluxes. Is this the aim?

**R** (P62): The reviewer is right. The DO flux at the coast should be represented by back and forth arrows. This has been corrected. (see Fig. A4).

The green line corresponds to the position of the 45µM mean DO isopleth, calculated at 25°S.

Although some mean features of the OMZ (the mean position of the 45 µM isopleth at 25°S, and the mean DO concentration at 100 m depth) are represented in the Figure 17 (new Figure 20), the aim of this figure is to synthesize the processes that intervene in the seasonal variability of the OMZ. The mean DO field was included only as a reference for the OMZ mean shape and position.

We added to the figure the modifications suggested by the reviewer concerning the representation of the DO flux from the coast, as well as the "Depth" axis label (Fig. A4). The figure caption was also improved. This figure corresponds to Figure 20 in the revised version of the manuscript.

[Figure]

**Figure A4.** Schematic of the main processes driving the seasonal variability in the SEP OMZ: The DO eddy flux through the northern-southern boundaries and the DO flux that takes place at the coastal boundary of the OMZ. The coastal band limits are defined by the light blue shading adjacent to the coast. A scale of the seasonal amplitude of the eddy driven DO flux at each OMZ boundary is indicated (units in cm s$^{-1}$ µM). The mean DO concentration (color shading) and the position of the 45 µM isopleth (thick black contour) at 100 m depth are also represented. The vertical/offshore DO flux induced by the propagation of the annual ETRW across the 45µM isopleth at 25°S is represented in the bottom panel.

**Response to Anonymous Referee #2**

We would like to thank the anonymous reviewer for his/her detailed review and helpful comments. As suggested by the reviewer, we have modified the current organization of the paper, improving the model validation (including an EKE comparison between the simulation and the available data and a Taylor diagram; Section 2.2) and separating the discussion from the conclusions.

**1. Response to major comments**

**Comment 1 (C1):** The additional section regarding the validation of the mesoscale activity within the model seems critical to identifying this mechanism. Specifically, the validation referenced in Gutknecht et al. 2013 seems to focus on validating the mean conditions, and Figure 6 of that work indicates some clear subsurface biases that need to be taken into account for these mechanisms. How do comparisons to the AVISO data set look? What about temperature? Can you rule out advection and confirm that the bias is in the biogeochemical model? The bias needs to be put in context –if the authors can describe the bias and then put their results in context that would provide much more information to the reader. The validation provided here is not only qualitative, but it also focuses on the average conditions. Some indication at to how well the model does at representing the variability and fluxes is necessary. Fluxes can include the SMS terms – like production. It is common now for taylor or target diagrams to be used to visualize the metrics of skill, this work would benefit from one that may also include some physical terms and goes beyond the climatology.

**Response (R.):** We agree with the reviewer that it is important to validate the simulation as much as we can, not only in terms of mean state but also in terms of seasonal variability. This model configuration has been validated from satellite and in situ observations in a previous study for the physical component (Dewitte et al., 2012). In particular the model exhibits a rather realistic mean SST (see figure A1) and mean thermocline along the coast (validated from the IMARPE cruise data, see figure 9 in Dewitte et al. (2012)) and the model mean EKE is comparable to the available observations (Fig. A2), although in general more intense, consistently with other model simulations in the region (see for example Fig. 5 in Colas et al. (2012)). The mean circulation near the coast is also in agreement with former modeling studies (See figure 3f of Dewitte et al. (2012); Montes et al., 2010). While Dewitte et al. (2012) focused on the validation of the circulation at interannual and decadal timescales, we provide in this paper material for assessing the realism of the seasonal variability, which consists in the Figure 5b (Figure 7b in the revised manuscript version) showing the comparison between CARS and model DO in terms of the seasonality of the volume distribution. In addition to providing further material for assessing the simulation, and following the reviewer's recommendation, we have also added a Taylor diagram to the revised version of the paper (Fig. A3), which complements the validation of the mean state and seasonal cycle inside the OMZ (Fig. 6). We obtain similar results to Montes et al. (2014) in terms of DO, temperature and salinity (see figure 1 in Montes et al., (2014)).

[Figure]

**Figure A1.** Mean sea surface temperature (SST) between 2000 and 2008 for (a) OISST product (0.25°x0.25°), (b) the simulation (1°/12) and (c) CARS dataset (0.5°x0.5°). (d) Difference between the OISST product and the simulation.

[Figure]

**Figure A2.** Mean Eddy Kinetic Energy (EKE) between 1993 and 2008 (satellite altimetry era), for (a) TOPEX/Poseidon Jason 1-2 merged product (0.25°x0.25°), and (b) Simulation (1°/12). EKE was derived from the interannual anomalies of the geostrophic velocity field.

[Figure]

**Figure A3.** Taylor diagram of the seasonal mean (hourglass, diamond, square and cross) and annual mean (circle) pattern of DO and Surface Chlorophyll (25°S-5°N, 88°W-70°W). Only annual mean pattern comparisons are shown for temperature and salinity (same spatial domain). DO, temperature and salinity were vertically averaged between 100 and 600m depth (focus on the OMZ core). Only the surface chlorophyll values within 250 km next to the coast were considered. The comparisons are made between the simulation and CARS (for DO, temperature and salinity) and SeaWiFS (for surface chlorophyll). Ordinate and abscissa axes represent the standard deviation normalized by the observations standard deviation. Blue dotted radial lines indicate the RMS difference between the observations and the simulation.

**C3:** The budget analysis is a powerful tool and has a lot of information within it. The authors appropriately use this tool to try and tease out the relative contribution of biological processes to the physical processes. It seems necessary to show the budget balances in the regions that this tool is used to discern the relative processes. A table of the budget terms would be one suggested way to achieve this goal. What does the climatological base budget look like? How big are the anomalies and residuals focused on in this work?

**R.** (P40): Following the reviewer's recommendation we provide in the revised manuscript a table (Table A1) that summarizes the seasonal anomalies in the DO budget inside the OMZ (comparing the Austral winter and summer values).

**Table A1.** Austral summer (DJF mean) and winter (JJA mean) seasonal anomalies of the DO budget, averaged over the core of the Peru Under Current at 12°S (as depicted by the red contour in Figure 12). The values for the seasonal cycle and the reconstructed first EOF mode (Figures 12 and 13) are presented along with the difference Climatology-EOF. All values are in $10^{-6}\mu M$ $s^{-1}$. Mixing here consists in the summep-up contribution of horizontal diffusion and ( $K_h \nabla^2 O_2$ ) and vertical diffusivity (

$\frac{\partial}{\partial z}\left(K_z \frac{\partial O_2}{\partial z}\right)$ ).

| | Climatology | | EOF | | Difference | |
| --- | --- | --- | --- | --- | --- | --- |
| | **Summer** | **Winter** | **Summer** | **Winter** | **Summer** | **Winter** |
| **dO$_2$/dt** | 1.10 | -2.74 | 1.30 | -2.67 | -0.2 | -0.07 |
| **Adv** | 0.61 | -9.38 | 0.85 | -9.30 | -0.24 | -0.08 |
| **Mixing** | -0.42 | 7.99 | -0.35 | 7.99 | -0.07 | 0.0 |
| **Biogeochemical Flux** | 0.91 | -1.35 | 1.00 | -1.35 | -0.09 | 0.0 |

**C4:** The methods section ends with a paragraph that seems more like discussion – describing vertical model decomposition. This doesn't seem to be very well woven into the rest of the paper and comes off as a bit of a distraction or outlier.

**R.** (P13, L6): The reviewer is right. In fact the modal decomposition is only used for providing a range of values for the phase speed of the Extra-tropical Rossby wave consistent with the model stratification. The phase speed values are further used to draw possible trajectories of the WKB ray paths (Figures 8 and 14). Following the reviewer's suggestion, we have simplified this paragraph and complemented the text in the results section, when we described the related figures.

**C5:** Given the paper's premise that the high resolution study provides further information than a coarse simulation would, some comparison seems warranted showcasing that result.

**R.:** Since the focus of the study is on the eddy flux driving the seasonality of the OMZ boundaries, the use of a high-resolution model is a requirement considering that low-resolution models will not realistically simulate the eddy activity (in general too low in IPCC-class models). Here we could not compare the model outputs used as boundary conditions of our regional model (i.e. SODA) and the actual regional model simulation in terms of DO concentration and seasonal DO eddy flux, since SODA does not provide DO concentration and we have been using DO from CARS as boundary condition of the regional model. In fact we provide in the paper a qualitative comparison between global models and our simulation, based on the OMZ geographical overlapping index defined by Cabré et al. (2015), which indicates that our model simulation is rather realistic compared to global models. This has to be attributed not only to the model's ability to realistically resolve the mean upwelling thanks to its resolution, but it has to do also with the realism of the simulated turbulent flow. This is certainly something that would be worth quantifying through dedicated model experiments. However it is beyond the scope of the present paper.

**C1:** The main comment in all the figures from the model fields is about the white space – what does it indicate?

**R.:** The white space represents a region where no data is available. In order to avoid confusion, we added this information in each figure caption where this is present.

**C2:** Figure 1 seems to indicate the model does not achieve the same onshore/offshore gradients in oxygen around 10 deg S. This seems critical to some of the points in the paper about transport mechanisms and goes unaddressed in the text.

**R.:** The reviewer is right. This could be a model bias. Note however that this is still in the range of what is expected from the use of different ocean boundary conditions. In particular, Montes et al. (2014) compared two simulations with OBCs having a slightly different mean circulation near the equator, and found differences between the simulations that are comparable to the one between CARS and our simulation. In addition it should be pointed out again that CARS data set is built from data covering a different period than our simulation, which could also explain the differences. It is difficult here to demonstrate if the difference between model and data originate from model bias, decadal variability or sampling issue in the data set. This has been mentioned in the text of the revised manuscript (P9, L16-18).

**C3:** Figures 1-3 would benefit from additional dots on the figures to identify where the samples the made up the climatology from the observations were made. The reader could discern from that addition the errors in interpolation to errors in the model.

**R.:** As suggested by the reviewer, the spatial resolution was included on the margins of the mentioned figures (as dots), in order to avoid difficulties in the interpretation. Figures 1-3 correspond to Figures 2-4 in the revised version of the manuscript.

**C4:** Figure 4 would benefit from isobaths contours so the shelf region was highlighted.

**R.:** The figure (Fig. 5 in the revised version of the manuscript) was modified as suggested by the reviewer.

**C5:** Figure 5 seems to indicate the model is biased high – that the model underestimates the hypoxic volume. Is that because of advection, temperature, or the bio model? Or could it be the different time periods compared between the model and obs?

**R.:** As pointed out by the reviewer, the simulation underestimates the hypoxic volume by ~6% compared to CARS, although very similar discrepancies between model and data have been obtained by previous modeling studies (e.g. Montes et al., 2014). As suggested by the reviewer, this could be due to several factors, including the OBCs used in our experiment, as well as the different time periods used to compute the DO climatology. Also, the model does not consider the benthic exchanges (no sediment model) that would tend to consume oxygen near the coast at the bottom of the water column, and could also explain such deficiency. Following the reviewer recommendation, we further expanded on this point in order to contextualize our results (P25, L22).

**C6:** Figure 6 – this figure and discussion are out of context and need to be reorganized at a more appropriate time in the results. These numbers need to be put in context with the other important fluxes as you do in the budget. So this needs to follow the budget.

**R.** (P13, L21-26): The purpose of Figure 6 (Figure 8 in the revised version of the manuscript) is to illustrate the spatial heterogeneity of the seasonal DO signal across the OMZ (Fig. 8a and 8b), and to document the propagating characteristics at different depths (Fig. 8c and 8d). It aims at introducing the idea that the upper OMZ seasonal cycle is eddy driven while the lower OMZ seasonal cycle is influenced by the propagation of the annual ETRW (Section 4.2). We believe that it would be confusing if we present it after the budget. Still, we have clarified the text at the beginning of Section 3, so as to put the figure and discussion in context. The text was modified as follows:
"**3 Characteristics of the DO annual cycle**

While the annual signal is a conspicuous feature inside the region (Fig. 6), it could manifest differently across the OMZ. As a first step towards investigating processes driving the rate of DO change, it appears important to document the vertical structure variability of the DO annual cycle within the OMZ. The amplitude and phase of the annual harmonic of the model DO climatology is presented along a zonal section off central Peru (12°S, Fig. 8ab), where the OMZ core is extensive (Fig. 2)".

**C7:** Figure 8 a and b panels do not appear to be on the same scale. Refer to Equation 2 from the text in the caption.

**R.:** They are indeed not in the same scale because the variability of the physical flux tends to be much larger than the variability of the biogeochemical flux. The equation 2 is not used for this figure (Fig. 10 in the revised version of the manuscript), it is just the variability (RMS), which can be viewed here as a measure of the amplitude of the seasonal cycle.

**C8:** Figure 10 is confusing but important. Clarity to the reader would be achieved through a longer caption explaining the different colorbars in a-c and d-g, as well as the percentages – which don't add up. Why is the undercurrent identified? Is it referred to in the text? Are these all model results? What transect is this in the domain? Does the choice of the transect change the results? The text refers to this figure describes seasonality, but this figure just says climatology – is it a seasonal climatology? In that case, what time of year is shown?

**R.** (P52): Following the reviewer's recommendation, we have detailed and improved the caption. The figure was also modified (Figure 12 of the revised manuscript version). This figure displays the results of an EOF analysis performed on different climatological fields for a zonal section at 12°S. Each percentage indicates the variance explained by the first EOF mode, for each field, in order to ascertain that the mode that is presented accounts for a significant share of the variance, and that we are not describing some peculiarities of the climatological fields. Therefore, the percentages are not meant to be summed-up. The temporal variability of these EOF mode patterns is provided in Figure 13 of the revised version of the manuscript.
The signal represented corresponds to the seasonal cycle, or seasonal climatology (see also the reply to C3 in major comments)
In our interpretation of the results, the seasonal intensification (destabilization) of the coastal alongshore current plays a decisive role in the seasonal DO changes at the shelf, which is to a large extent associated with transport within the Peru Chile Undercurrent (Montes et al., 2010), that is why it is included in the panels of Figure 12 and mentioned several times in the text.

We choose to illustrate the seasonality of the OMZ with a section at 12°S given that it is located at the core of the OMZ. The results are insensitive to the choice of the latitude at which the EOF analysis is performed, for a given latitude between 7°S and 14°S, which corresponds to the latitude range where the PUC is well defined (See for instance the results for 9°S (Fig. A4)).

[Figure]

**Figure A4.** First EOF mode pattern of (a) DO, (b) Alongshore currents component, (c) Eddy Kinetic Energy, (d) oxygen rate, (e) biogeochemical flux, (f) advective terms (sum of horizontal and vertical components) and (g) mixing terms (sum of horizontal and vertical components). Percentage of explained variance by each EOF mode pattern is indicated in parentheses on top of each panel. The red contour denotes the mean position of the Peru Under Current core, defined here as along-shore southward current exceeding 4 cm s⁻¹. The black dashed contour denotes the mean DO 45µM isopleth. (h, i) Non dimensional principal components (PC) associated with the EOF patterns. RMS values are indicated between parentheses. The residual corresponds to the difference between the rate of DO change and the sum of all the terms of the rhs of Eq. 1 in terms of the normalized PC timeseries. The weak residual indicates that the seasonal DO budget can be interpreted from the EOF decomposition. The EOF decomposition was performed over the climatological (mean seasonal cycle) fields. Multiplying the EOF pattern by the corresponding PC timeseries yields the contribution of the first EOF mode to the original field, in dimensionalized units. Land and the region outside the 45 µM mean DO isopleth are masked in white (a-g).

**C9:** Figure 11 requires more explanation as to how it was made. No explanation of residuals etc is provided in the text. How should the reader interpret the residual? Is it meant to just be physical?

**R.** (P53): Following the reviewer's recommendation we have detailed the caption of Figure 11 (Fig. 13 in the revised manuscript version; see also C8). The residual is computed from the difference between the principal component of the rate of DO change and the summed-up contribution of the principal components corresponding to the physical and biogeochemical DO fluxes in terms of the normalized PC timeseries. This calculation is performed in order to verify that the DO budget based on the EOF decomposition is almost closed, which allows for its interpretation.

The residual should be interpreted as the "error" in approximating the terms of the DO budget by their first EOF component. This approximation has the advantage of synthesizing the DO budget, yielding (1) a spatial pattern that allows identifying where the seasonal variations are more important and (2) the phase of the seasonal maximum.

As suggested by the reviewer, the caption of the Figure 11 (now Fig. 13) was improved in order to clarify what the residual means: "
[revised manuscript text omitted]

---

## Author Response (AR2)

**Response to Anonymous Referee #2**

We would like to thank the reviewer for his/her constructive comments that helped to improve the manuscript. We have addressed all of them and provide below a detailed reply. Please note that the page and line numbers refer to the marked-up version of the revised manuscript.

**Comment 1 (C1; P. 10, L. 5-6):** "the modeled oxygen content is however underestimated as compared to CARS in certain regions of the domain, " This should be rephrased to clearly state the model is biased high in certain regions and underestimates the volume of suboxic water. This may be due to the vigorous mixing occurring in the EKE figure the authors included with their comments. Pointing to a reason for the bias would provide ways to shape the conclusions presented here.

**Response (R.):** Following the reviewer's recommendation, the sentence was reformulated so that it is clear that the model is biased high in certain regions of the domain (P10, L9-13): "...the simulation overestimates the oxygen content in certain regions of the domain as compared to CARS, particularly southwards of 20°S (Fig. 3a) and close to the coast (Fig. 3d). The simulation also underestimates by 6% the volume of suboxic water (Fig. 6a), which is comparable to the differences obtained by Montes et al. (2014) using the same model within a different configuration and boundary forcing".

We also agree with the reviewer on that is important to contextualize our results in terms of the influence that the biases in the mesoscale activity levels could have on the simulated suboxic water volume. We further expanded on this point in the discussion section (P22, L23).

**C2 (Page 10, Line 13-14):** The observations are not the only thing at fault. No model is perfect and so the biases here should be openly discussed. Please make sure to discuss the results with an even hand.

**R.:** The sentence was reformulated as (P10, L18-23): "...In particular, the model oxycline is shallower and with a more intense DO gradient than the observations, which has been also observed in a regional simulation of the Arabian Sea OMZ (Resplandy et al., 2012). While this could be partly due to CARS underestimating the DO gradient, as a result of its relatively low vertical resolution, it could also be that the model underestimates the vertical diffusivity in the vicinity of the oxycline".

**C3 (Page 22, Lines 8-10):** Bianucci et al is not the only oxygen model in the NCCS which investigated the role of sediments. Siedlecki et al (2015) found them to play a significant role on the Washington shelf, and they have been found to be important in other regions as well.

**R.:** We thank the reviewer for pointing out this reference that we have included in the revised manuscript when we discuss the deficiency of the model in simulating low DO concentration near the shelf (P22, L18). That part of the paragraph was reformulated as follows:

"The role of benthic processes in constraining the DO demand has been studied in the northern

California Current system (Bianucci et al., 2012; Siedlecki et al., 2015) indicating that locally, such processes might be essential to explain the hypoxic conditions. The inclusion of a sediment module in the current model setting is planned for future work to address this issue".

**C4 (Page 22, Lines 8-10):** While I applaud the authors for trying to present the limitations of the model results prior to the discussion of implications, it seems prudent here to discuss the results from the figure included in your comments to reviewers (figure A2 or Figure 1 in the paper). Specifically, the simulation seems quite a lot higher in EKE south of 20 S. so the "ventilation" and EKE contributions in this region of the budget should be viewed within the context of this bias. The entire discussion of the Southern Boundary and the mechanisms surrounding that region need to be backed off and put in the context of the mixing and probably ventilation in the region being biased high.

**R.:** We agree with the reviewer. We expanded the discussion around the possible impact that biases in the mesoscale activity levels could have on the DO distribution in the simulation (P22, L23). "Another process that could contribute to the underestimation of the suboxic volume in the simulation is the higher mesoscale activity in the model compared to the observations (Fig. 1) that likely participates to ventilate the OMZ more than in nature".

As suggested by the reviewer, the discussion of the mechanisms driving the eddy activity in the southern boundary was also rewritten (P24, L20).

"A possible mechanism driving the local variability observed at the southern section is the generation of local baroclinic instability and vorticity input from wind stress curl as observed for the California system (Kelly et al, 1998). The southern section lies within the northeast rim of the Southeast Pacific Anticyclone, and the peak in the seasonal DO eddy flux coincides with the reported intensity peak of the seasonal cycle of the Anticyclone, towards the end of the year (Rahn et al., 2015; Ancapichún and Garcés-Vargas, 2015). The mesoscale activity in this region could be directly modulated by the winds. An additional source of intraseasonal (internal) variability in the currents field could be the interaction between the annual extra tropical Rossby wave and the mean circulation (Dewitte et al., 2008; Qiu et al., 2013). The actual source of the eddy activity in this region would also deserve further investigation."

**C5:** That said, a lot of other results in the paper are worthy of highlighting, but the boundary stories seems like the wrong one to focus on in areas of the discussion and conclusions. Altering the presentation of this result in the paper should take minor revisions.

**R.:** Following the reviewer's recommendation, we alleviated the focus on the northern, southern and coastal boundaries in the discussion section (P22-26). However, we consider that it is important to highlight the different dynamical regimes and seasonality of these boundaries, which provides a paradigm for interpreting its variability at longer timescale.

[revised manuscript text omitted]